# TOWARD ROBUST FEATURE SPACE IN LONG-TAILED TIME SERIES CLASSIFICATION: A MULTI-SCALE PERSPECTIVE

## ABSTRACT

In recent years, time-series classification (TSC) has seen significant progress. Nevertheless, research on long-tailed TSC remains relatively limited. A key issue in long-tailed scenarios is that high inter-class similarity often leads models to learn overlapping features, making tail classes particularly difficult to distinguish. This phenomenon gives rise to three specific challenges: (1) Conventional approaches based on oversampling or uniform-intensity data augmentation may overfit or fail to learn robust features for tail classes. (2) Limited model representation capacity can lead to aligned temporal features across classes, further exacerbating class confusion. (3) Such class overlap makes it challenging to establish discriminative decision boundaries, particularly in highly imbalanced scenarios. To address these challenges, we propose TimeLT, a novel framework designed to learn a robust and discriminative feature space from long-tailed time-series data. First, we introduce a personalized augmentation strategy that generates tailored perturbations for scarce tail samples, preventing overfitting while increasing sample diversity. Second, we employ a multi-scale temporal encoder to capture patterns at different temporal resolutions, enabling the model to extract informative and discriminative features for both head and tail classes. Third, we propose a boundary-repelling regularization term that encourages embeddings to move closer to their respective class centroids while being repelled from inter-class boundaries, promoting compact and well-separated feature representations. To promote comprehensive research in this area, we consolidate a dedicated benchmark comprising several long-tailed datasets and over 16 advanced baselines. Extensive experiments across all datasets demonstrate that TimeLT significantly outperforms the strongest baselines, achieving accuracy improvements ranging from 0.55% to 12.27%. The anonymized source code is available at `https://anonymous.4open.science/r/TimeLT-A11C`.

## 1 INTRODUCTION

Advancements in deep learning have given rise to neural network-based time series classification (TSC) architectures capable of capturing rich temporal patterns and significantly improving classification accuracy (Auger-Méthé et al., 2021; Morid et al., 2023; Wang et al., 2023a). However, these time-series classification models frequently contend with long-tail distributions in which a few head classes dominate sample counts while many classes are severely underrepresented (Zhao et al., 2022). This skew distorts the training process and shifts decision boundaries toward the head classes, diminishing the model's ability to learn from rare categories. The problem is pervasive in real-world settings, for example, rare but critical diseases in medicine or infrequent extreme pollution events in environmental monitoring. As Figure 1 shows, extreme pollution represents only 4.26% of observations across ten monitoring sites in China. Such imbalance erodes performance on scarce yet important classes, potentially producing serious practical consequences.

Inspired by recent advances in long-tail learning in computer vision, researchers have begun adapting these techniques to long-tailed TSC. For example, FSR (Wang et al., 2023c) and COCL (Wu et al., 2025) proposes a supervised contrastive objective that aligns samples with class prototypes at multiple granularities, mitigating representation collapse for underrepresented classes. Similarly,

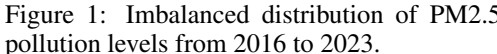

Figure 1: Imbalanced distribution of PM2.5 pollution levels from 2016 to 2023.

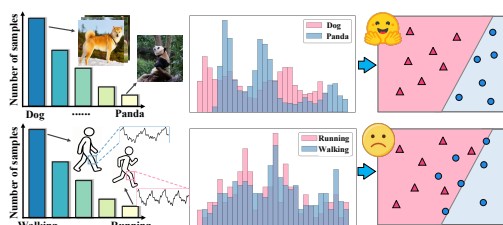

Figure 2: Comparison of data distributions in computer vision (upper) and time series (low).

DGMSCL (Qian et al., 2025) combines dynamic graph learning techniques with a context-based minority-class-aware contrastive loss.

*Although these studies demonstrate the strong adaptability of computer vision techniques to TSC, they overlook the unique characteristics of time series data compared to image data.* Time series data often exhibit higher inter-class similarity, which makes it more challenging for classifiers to make accurate predictions. Specifically, in the image domain, as shown at the top of Figure 2, head and tail classes (e.g., dogs and pandas) typically have distinct data distributions, enabling models to learn highly separable representational spaces. In contrast, for time series data, as shown at the bottom of Figure 2, activities like "walking" and "running" represent head and tail classes, respectively. Knee and foot sensor signals display highly similar temporal patterns, creating inter-class redundancy that impedes the model's ability to learn distinctive, robust features for each activity. To further substantiate, we computed the JS divergence between class-wise distributions on the canonical image dataset CIFAR-LT and our HAR-LT and Epilepsy datasets, as shown in Figure 3. CIFAR-LT exhibits larger divergence values, indicating clearer inter-class separability. In contrast, HAR-LT and Epilepsy display smaller divergence values, highlighting the higher inter-class similarity inherent in time series data.

The unique characteristics of time series data introduce specific challenges for long-tailed time series classification. First, the high similarity between classes causes existing methods that rely on oversampling or uniform-intensity data augmentation to overfit or fail to learn robust features for tail classes. Second, when the model's representation capacity is limited, this similarity leads to aligned temporal features. Thus, such lass overlap makes it more difficult for the model to learn discriminative decision boundaries in long-tailed scenarios (Fachrie et al., 2025). As a result, predictions for tail classes become confused and their accuracy declines substantially.

To overcome these challenges, we propose **TimeLT**, a long-tail classification framework for time series that focuses on extracting discriminative inter-class features and forming clearer decision boundaries to boost accuracy on underrepresented classes. First, TimeLT applies personalized augmentation that tailors perturbations to each class. This reduces overfitting on rare classes and increases input-level separability. Moreover, a multi-scale temporal encoder captures patterns across multiple time resolutions, mitigating interference from redundant inter-class signals and producing richer, more informative representations. To shape the representation space, we introduce a boundary-repelling regularizer that pulls intra-class embeddings toward their class centroids while pushing them away from nearby decision boundaries. The combined effect yields a more balanced and separable feature space that enhances tail-class discrimination. Finally, we will release a comprehensive long-tail time-series benchmark that aggregates real-world datasets, provides standardized splits, and includes baseline implementations (detailed information is provided in Appendix B). The benchmark will be maintained and expanded to support ongoing research. Our contributions are summarized as follows:

❶ *Practical Exploration.* To accelerate research on long-tailed TSC, we release a comprehensive benchmark that standardizes data processing protocols, provides a diverse set of datasets and long-tail splits, and includes strong baseline implementations. The benchmark enables fair, reproducible, and holistic evaluation across a variety of long-tail scenarios.

❷ *Unbiased Temporal Encoding.* TimeLT incorporates a customized representation learning module tailored for long-tailed time series. This strategy integrates oversampling, data

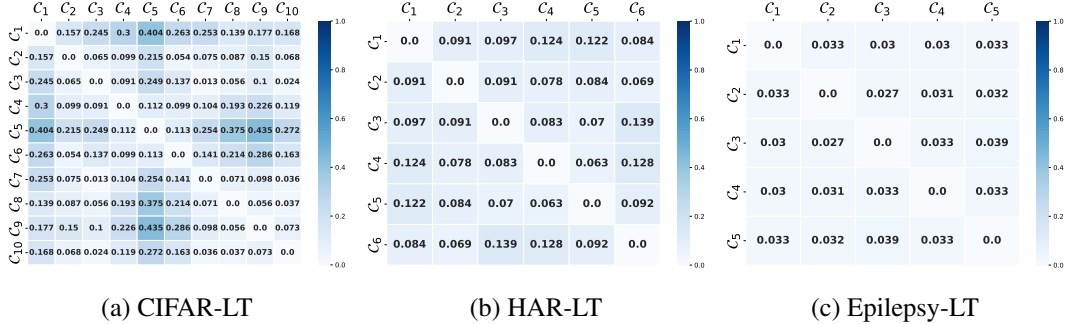

(a) CIFAR-LT         (b) HAR-LT         (c) Epilepsy-LT

Figure 3: JS divergence matrices for class-wise distributions on different datasets: CIFAR-LT (image dataset), and HAR-LT and Epilepsy-LT (time series datasets).

- ❸ *Discriminative Optimization Objective.* TimeLT includes boundary-repelling and directional regularization terms, which explicitly repel sample embeddings from the decision boundary toward their class centroids, enforcing a more separable and robust feature space.

- ❹ *Empirical Study.* Extensive experiments demonstrate that our proposed TimeLT achieves state-of-the-art performance. Compared with the strongest baselines across all datasets, TimeLT achieves overall accuracy improvements ranging from 0.55% to 12.27%.

## 2 RELATED WORK

**Time Series Analysis**   Deep learning has achieved substantial success in time series representation learning, owing to its ability to capture informative temporal representations (Trirat et al., 2024). These representations can be applied to various downstream tasks, such as classification, forecasting, and imputation. Current research has leveraged a range of deep architectures, including Convolutional Neural Networks (CNNs) (Wu et al., 2022; Wang et al., 2023b; Mu et al., 2025), Recurrent Neural Networks (RNNs) (Lin et al., 2023; Beck et al., 2024), and Graph Neural Networks (GNNs) (Wang et al., 2024a;f), often enhanced with specialized components to capture complex patterns such as periodicity, trends, interactions among variables, and frequency-domain structures (Wu et al., 2021; Peng et al., 2024). More recently, Transformer- and MLP-based models have gained momentum in time series analysis due to their strong representation capacity and scalability (Nie et al., 2022; Wang et al., 2024d; Le et al., 2024; Stitsyuk & Choi, 2025). However, these methods primarily focus on architectural design to capture temporal dependencies, while overlooking the challenges posed by long-tailed data distributions.

**Long-tail Learning for Time Series**   Early work on long-tailed time series classification (TSC) emphasized data-level remedies such as oversampling and undersampling. For instance, INOS (Cao et al., 2013) synthesizes minority-class instances by sampling from multivariate Gaussian models, while T-SMOTE (Zhao et al., 2022) leverages an auxiliary LSTM-based classifier to score minority examples and generates new time subseries using varying prefix lengths. In parallel, methods inspired by the image domain adopted contrastive learning to improve class separability (Wang et al., 2023c; Wu et al., 2025); COCL (Wu et al., 2025), for example, introduces a contrastive clustering loss that reduces intra-class variance and enlarges inter-class margins. DGMSCL takes a different route by directly amplifying the learning weights for underrepresented classes (Qian et al., 2025). Despite these advances, many approaches fail to address the high inter-class similarity intrinsic to time series data, which impedes the formation of compact, well-separated feature manifolds, an issue that is exacerbated under extreme class imbalance.

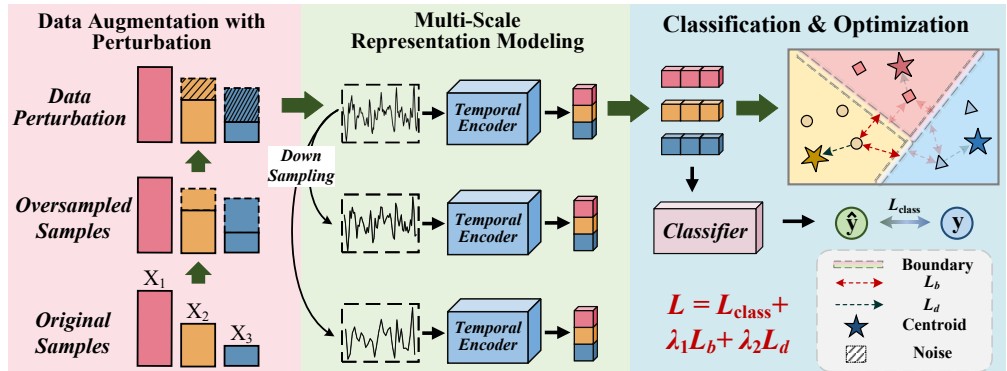

Figure 4: The framework of TimeLT.

## 3 PRELIMINARY

**Long-tailed Time Series Classification** Given a time series training set $\mathcal{D} = \{\mathbf{X}_n, \mathbf{Y}_n\}_{n=1}^N$, where $\mathbf{X}_{n \in N} \in \mathbb{R}^{T \times D}$ denotes the $n$-th time series sample with length of $T$ and $D$ variables, $N$ is the total number of samples, and $\mathbf{Y}_{n \in N} \in \{0, 1\}^C$ is the corresponding label annotation, where $C$ means the number of classes. Let $N_c$ denote the number of samples in the $c$-th class, where the class indices are sorted in descending order (i.e., if $i < j$, $N_i \geq N_j$). We also denote $\mathcal{C} = \{\mathcal{C}_c\}_{c=1}^C$ as the set of $C$ class labels. However, in real-world applications, the number of samples across classes is typically highly imbalanced, exhibiting a long-tailed distribution, (i.e., $N_1 \gg N_C$), and the imbalance ratio (IR) is defined as $\frac{N_1}{N_C}$. The objective of the task is to learn a temporal encoder $f : \mathbf{X} \rightarrow \mathbf{z}$, and a classifier $\phi : \mathbf{z} \rightarrow \widehat{\mathbf{Y}}$ for producing valid label predictions.

**Definition 1.** *Class Centroid. The centroid of a class is defined as the mean of all feature embeddings belonging to that class. Formally, the centroid $\mu_c$ of $\mathcal{C}_c$ is formulated as: $\mu_c = \frac{1}{N_c} \sum_{n=1}^{N_c} \mathbf{z}_n$.*

**Definition 2.** *Decision Boundary. The decision boundary refers to the hypersurface in the feature space that separates feature embeddings of different classes. For two class-balanced feature spaces, the distances from the decision boundary $\mathbf{b}_{a,b}$ to the centroids of $\mathcal{C}_a$ and $\mathcal{C}_b$ are equal, which can be expressed as: $\|\mathbf{b}_{a,b} - \mu_a\|_2 = \|\mathbf{b}_{a,b} - \mu_b\|_2$. However, in practice, feature distributions are rarely perfectly balanced. We introduce a scaling factor based on the standard deviation $\sigma$ of each class: $\frac{\|\mathbf{b}_{a,b} - \mu_a\|_2}{\sigma_a} = \frac{\|\mathbf{b}_{a,b} - \mu_b\|_2}{\sigma_b}$.*

## 4 METHOD

### 4.1 OVERALL ARCHITECTURE

Figure 4 shows the overall architecture of the proposed TimeLT. ❶ In the **data preprocessing stage**, a personalized perturbation-based data augmentation strategy is designed to balance the feature distributions across different classes and enhance data diversity. ❷ In the **time series representation learning phase**, TimeLT consists of the following steps: (1) A multi-scale temporal encoder captures rich temporal patterns from input time series and maps them into a discriminative feature space. The multi-scale temporal features helps improve generalization, particularly for tail classes; (2) The learned temporal representations are then fed into a classifier for generating prediction. ❸ During **the loss optimization phase**, TimeLT incorporates two main objectives: (1) Minimizing the discrepancy between predictions and ground-truth labels using a cross-entropy loss; (2) The two regularization terms encourage samples to move away from shared decision boundaries of other classes and toward their respective class centroids, thereby promoting a discriminative decision boundary that is robust to class imbalance.

## 4.2 MULTI-SCALE ENHANCEMENT MODELING FOR TIME SERIES REPRESENTATION

### 4.2.1 PERSONALIZED DATA AUGMENTATION WITH PERTURBATION

Techniques such as jittering, cropping, and warping are widely used for data augmentation. However, uniform augmentation becomes problematic under severe imbalance, potentially leading to feature overlapping and overfitting in tail classes (Qian et al., 2025). In this paper, we propose a novel data augmentation approach, which contains two phases and applies class-wise personalized augmentation to enhance data diversity. Specifically, for any class $c \in \{1, \cdots, C\}$, we first calculate the quantity of augmented time series samples for expansion of this class:

$$M_c = \lfloor (N_1 - N_c) \cdot \beta \rfloor, \tag{1}$$

where hyperparameter $\beta \in [0, 1]$ is the quantity balance factor, controlling the number of oversampled samples. $N_1$ represents the sample number of the most abundant class. Then, we randomly select $M_c$ samples in the sample set of category $c$, and the output is denoted as $\mathcal{X}_c^a = \{\mathbf{X}_m^{a,c}\}_{m=1}^{M_c}$. This inverse imbalance sampling method effectively enlarges tail classes.

However, directly using these oversampled samples in training may introduce feature redundancy and lead to overfitting. To enhance data diversity and improve generalization, we further apply a data perturbation strategy that introduces controlled stochasticity by gradually adding Gaussian noise with increasing intensity from majority to tail classes. For any oversampled sample $\mathbf{X}_n^{a,c}$ in $\mathcal{X}_c^a$, we add the noise $\epsilon$,

$$\mathbf{X}_n^{aug,c} = \mathbf{X}_n^{a,c} + \epsilon, \quad \epsilon \sim \mathcal{N}(0, \sigma_c^2 I), \tag{2}$$

where the noise factor $\sigma_c$ is class-dependent and defined as:

$$\sigma_c = \alpha \times \mathcal{D}_{train}(\text{std}) \times \frac{N_C - N_c}{N_C}, \tag{3}$$

where $\mathcal{D}_{train}(\text{std})$ represents the standard deviation of time series samples in the training set. The hyperparameter $\alpha$ controls the degree to which augmentation adapts to class imbalance.

Therefore, the enhanced training set can be constructed as: $\mathcal{D}_{train} = \{\mathbf{X}_n, \mathbf{y}_n\}_{n=1}^N \cup \{\mathbf{X}_m^{aug}, \mathbf{y}_m^{aug}\}_{m=1}^M$. For simplicity, the total number of samples in $\mathcal{D}_{train}$ is redefined as $N$.

### 4.2.2 MULTI-SCALE TEMPORAL ENCODER

We employ a simple but surprisingly effective temporal encoder, which integrates multi-scale analysis with a channel-independent strategy to capture rich multi-scale temporal patterns.

**Multi-scale Series Generation** Multi-scale analysis approaches have been proven effective in time series analysis, as these methods capture more comprehensive temporal patterns (Mozer, 1991). For example, fine-grained original input series can reflect detailed temporal variations, whereas downsampled series exhibit coarser patterns, such as trend information. Following previous studies, the historical input series $\mathbf{X}_{n \in N}$ is sampled at different temporal resolutions, generating a set of $S$-scale time series denoted as $\mathcal{X}_n = \{\mathbf{X}_n^s\}_{s=1}^S$. Specifically, $\forall \mathbf{X}_n^{s \in S} \in \mathbb{R}^{\lceil \frac{T}{s} \rceil}$ is obtained as:

$$\mathbf{X}_n^{s \in S} = \text{DownSamping}(\mathbf{X}, \text{kernel} = s, \text{stride} = s), \tag{4}$$

where $s \in \{2^{s-1}\}_{s=1}^S$, DownSamping$(\cdot)$ can be implemented using average pooling, max pooling, or convolution, and $\mathbf{X}_n^1 = \mathbf{X}_n$ represent the original input sample.

**Channel Independence with 1D-CNN** The channel-independent learning strategy avoids redundant feature learning across dimensions (Nie et al., 2022), thereby enhancing the model's expressiveness, particularly for tail classes. To capture local temporal patterns at each scale while preserving temporal structure, we employ a 1D-CNN module,

$$\text{for } s \text{ in } [1, \cdots, S] : \mathbf{H}_n^s = \text{Embedding}_{(s)}(\mathbf{X}_n^s), \tag{5}$$

where $\mathbf{H}_n^s \in \mathbb{R}^{T_s \times D_{model}}$ is the high-level representations.

**Global Temporal Modeling** Subsequently, we employ a two-layer GRU network to model global temporal dependency across $S$ scales, which captures sequential dependencies through an ordered learning mechanism. Here, we choose GRU over Transformer-based architectures due to its higher computational efficiency and its ability to effectively model sequential dependencies through memory-based mechanisms. The comparison is provided in Table 4. Specifically, the temporal representation of the $s$-th scale $\mathbf{z}_n^s$ can be generated as:

$$\text{for } s \text{ in } [1, \cdots, S] : \mathbf{z}_n^s = \text{GRU}(\mathbf{H}_n^s) \in \mathbb{R}^F. \tag{6}$$

After modeling each scale, we can obtain the output representation $\mathbf{z}_n = \overset{S}{\underset{s=1}{\|}} \mathbf{z}_n^s$, where $\|$ is the concatenation operation. Finally, $\mathbf{z}_n$ is fed into a two-layer feed-forward network (FFN) for predicting class labels $\widehat{\mathbf{Y}}_n \in \mathbb{R}^C$.

### 4.3 DECISION BOUNDARY REPULSION REGULARIZATION

In classification tasks, long-tailed class distributions often lead to decision boundaries that are dominated by head classes, resulting in feature space overlap between head and tail classes, particularly under extreme class imbalance. This issue is especially pronounced in long-tail TSC, where certain time series or their subsequences may exhibit similar temporal patterns across different categories. Such similarity can confuse the classifier, causing tail-class instances to be mapped near shared decision boundaries, thereby increasing the risk of misclassification. To address this, we propose a boundary-repelling regularization term that explicitly repels samples away from decision boundaries that are adjacent to other classes in the feature space. Specifically, for the output representation of $i$-th category $\forall \mathbf{z}_i \in \mathcal{C}_i$,

$$\mathcal{L}_b(\mathbf{z}_i) = \frac{1}{N_i - 1} \sum_{\mathbf{b}_{i,j} \in \mathcal{B}_i} Sim\left(\mathbf{z}_i, \mathbf{b}_{i,j \neq i}\right), \tag{7}$$

where $\mathcal{B}_i = \{\mathbf{b}_{i,1}, \mathbf{b}_{i,2}, \ldots, \mathbf{b}_{i,C}\} \setminus \mathbf{b}_{i,i}$ denotes the set of decision boundary embeddings corresponding to the classes whose feature spaces are adjacent to that of $\mathcal{C}_i$. $Sim(\cdot, \cdot)$ denotes the cosine similarity. Moreover, to ensure the correctness of the repulsion direction, we employ the supervised contrastive loss contrasting to pull temporal embeddings closer to the centroid of their corresponding class,

$$\mathcal{L}_d(\mathbf{z}_i) = -\log \frac{\exp\left(Sim\left(\mathbf{z}_i, \mu_i\right)\right)}{\sum_{\mu_j, j \neq i} \exp\left(Sim\left(\mathbf{z}_i, \mu_j\right)\right)}, \tag{8}$$

where $\mu_i$ is the centroid of $\mathcal{C}_i$ to which $\mathbf{z}_i$ belongs, as defined in Definition 1. Finally, the training objective can be formulated as,

$$\mathcal{L} = \mathcal{L}_{class} + \lambda_1 \mathcal{L}_b + \lambda_2 \mathcal{L}_d, \tag{9}$$

where $\mathcal{L}_{class}$ is the classification loss (i.e., cross-entropy). $\mathcal{L}_b$ and $\mathcal{L}_d$ encourage each sample to move away from the decision boundary while drawing closer to its corresponding class centroid. This leads to more compact and well-separated class distributions in the representation space, resulting in a more discriminative decision boundary. $\lambda_1$ and $\lambda_2$ are the hyperparameters striking a balance between three terms.

## 5 EXPERIMENT

### 5.1 LONG-TAIL TIME SERIES CLASSIFICATION BENCHMARK

**Datasets** We evaluated TimeLT across four datasets under three levels of class imbalance: low (IR = 10), medium (IR = 50), and extreme (IR = 100). Three datasets were adapted from established benchmarks, Human Activity Recognition (HAR), Sleep Stage Classification (ISRUC), and Epileptic Seizure Detection (Epilepsy) (Eldele et al., 2021; Wang et al., 2024g), and served to test performance on commonly studied time-series tasks. In addition, we assembled a five-year PM2.5 dataset from ten air quality monitoring stations. Samples were labeled into three categories, no pollution, moderate pollution, and severe pollution, and exhibit naturally imbalanced class frequencies (see Figure 1). Further dataset descriptions and preprocessing details are provided in Appendix B.1.

Table 1: Average performance over five runs. **Bold** & Underline indicate the best & second best results, respectively. * denotes that the improvement over the second-best results is statistically significant according to a t-test with $p < 0.01$. IR represents the imbalance ratio. The HAR-LT, ISRUC-LT, and Epilepsy-LT datasets follow a long-tailed (exponential) distribution. The remaining results are presented in Table 7 of the Appendix.

| Methods | Metrics | HAR-LT | | | ISRUC-LT | | | Epilepsy-LT | | | PM2.5 |
|---|---|---|---|---|---|---|---|---|---|---|---|
| | | IR=10 | IR=50 | IR=100 | IR=10 | IR=50 | IR=100 | IR=10 | IR=50 | IR=100 | IR=11.17 |
| DLinear [AAAI 2023] | ACC(%) | 56.75 | 50.84 | 49.61 | 25.43 | 24.83 | 23.72 | 20.39 | 20.39 | 20.39 | 78.42 |
| | F1(%) | 53.70 | 44.82 | 41.54 | 20.79 | 19.56 | 18.14 | 6.77 | 6.77 | 6.77 | 50.87 |
| | MCC | 0.500 | 0.467 | 0.458 | 0.019 | 0.006 | -0.008 | 0.000 | 0.000 | 0.000 | 0.474 |
| PatchTST [ICLR 2023] | ACC(%) | 83.95 | 75.85 | 74.02 | 63.32 | 59.08 | 53.14 | 60.62 | 49.92 | 43.14 | 73.37 |
| | F1(%) | 83.87 | 74.45 | 72.47 | 59.36 | 49.35 | 43.75 | 57.81 | 46.48 | 39.18 | 47.43 |
| | MCC | 0.809 | 0.717 | 0.699 | 0.544 | 0.488 | 0.400 | 0.532 | 0.417 | 0.347 | 0.410 |
| xPatch [AAAI 2025] | ACC(%) | 89.51 | 77.48 | 72.08 | 19.73 | 19.73 | 19.73 | 45.37 | 20.39 | 20.39 | 76.51 |
| | F1(%) | 89.20 | 76.49 | 69.85 | 6.59 | 6.59 | 6.59 | 37.83 | 6.77 | 6.77 | 50.11 |
| | MCC | 0.874 | 0.731 | 0.671 | 0.000 | 0.000 | 0.000 | 0.381 | 0.000 | 0.000 | 0.470 |
| Timemixer++ [ICLR 2025] | ACC(%) | 90.52 | 73.61 | 69.63 | 56.77 | 47.48 | 42.20 | 20.39 | 20.39 | 20.39 | 76.61 |
| | F1(%) | 90.37 | 71.52 | 65.45 | 46.61 | 31.21 | 24.19 | 6.77 | 6.77 | 6.77 | 49.33 |
| | MCC | 0.888 | 0.691 | 0.652 | 0.462 | 0.329 | 0.230 | 0.000 | 0.000 | 0.000 | 0.470 |
| PatchMLP [AAAI 2025] | ACC(%) | 90.93 | 75.54 | 68.98 | 51.57 | 41.75 | 38.66 | 20.39 | 20.39 | 20.39 | 78.57 |
| | F1(%) | 90.81 | 74.33 | 65.07 | 42.10 | 27.92 | 23.46 | 6.77 | 6.77 | 6.77 | 50.38 |
| | MCC | 0.891 | 0.711 | 0.643 | 0.373 | 0.246 | 0.184 | 0.000 | 0.000 | 0.000 | 0.483 |
| MPTSNet [AAAI 2025] | ACC(%) | 90.65 | 78.83 | 76.87 | 65.48 | 56.64 | 53.96 | 51.02 | 45.20 | 42.65 | 75.96 |
| | F1(%) | 90.51 | 77.18 | 74.40 | 59.70 | 46.66 | 39.66 | 45.16 | 36.17 | 34.22 | 50.37 |
| | MCC | 0.889 | 0.754 | 0.731 | 0.551 | 0.446 | 0.411 | 0.414 | 0.360 | 0.328 | 0.493 |
| CFAMG [KDD 2025] | ACC(%) | 90.12 | 74.83 | 72.13 | 34.28 | 32.60 | 30.33 | 21.91 | 20.39 | 20.39 | 78.43 |
| | F1(%) | 90.03 | 73.09 | 69.77 | 28.49 | 25.56 | 24.12 | 11.89 | 6.78 | 6.78 | 52.04 |
| | MCC | 0.882 | 0.705 | 0.677 | 0.125 | 0.095 | 0.062 | 0.030 | 0.000 | 0.000 | 0.497 |
| DGMSCL [NN 2025] | ACC(%) | 92.20 | 86.04 | 86.36 | 82.22 | 65.19 | 58.73 | 43.97 | 37.39 | 35.48 | 78.75 |
| | F1(%) | 92.06 | 85.56 | 85.92 | 78.08 | 50.68 | 42.15 | 35.80 | 29.70 | 27.42 | 52.63 |
| | MCC | 0.907 | 0.836 | 0.839 | 0.763 | 0.578 | 0.523 | 0.376 | 0.294 | 0.255 | 0.508 |
| TimeLT (Ours) | ACC(%) | **97.41***  | **94.99*** | **92.01*** | **82.77*** | **77.71*** | **73.12*** | **65.81*** | **61.04*** | **55.41*** | **83.24*** |
| | F1(%) | **97.38*** | **94.83*** | **91.60*** | **80.29*** | **75.77*** | **70.28*** | **64.84*** | **60.84*** | **54.46*** | **61.26*** |
| | MCC | **0.969*** | **0.940*** | **0.905*** | **0.780*** | **0.714*** | **0.655*** | **0.582*** | **0.520*** | **0.453*** | **0.584*** |

**Evaluation Metrics** We use accuracy (ACC), F1-score (F1), and Matthews correlation coefficient (MCC) to assess the performance. MCC can provide a more comprehensive performance evaluation on imbalanced datasets. Its value ranges between $[-1, 1]$, where MCC $= 0$ indicates random predictions, and MCC $= -1$ implies predictions that are completely opposite to the observed labels (Boughorbel et al., 2017). For more details, please refer to Appendix B.2.

**Baseline Models** Our benchmark comprises 16 state-of-the-art time-series models selected to cover a wide range of downstream tasks. We organize these models into two categories: traditional time-series analysis methods and approaches specifically designed to handle long-tail distributions in time-series learning. ❶ **Traditional Time-series Analysis Methods** include DLinear (Zeng et al., 2023), PatchTST (Nie et al., 2022), xPatch (Stitsyuk & Choi, 2025), TimeMixer++ (Wang et al., 2024c), PatchMLP (Kong et al., 2025) and MPTSNet (Mu et al., 2025). ❷ **Long-tailed Time-series Classification Methods** include DGMSCL (Qian et al., 2025) and CFAMG (Wang et al., 2025). Detailed introduction are provided in Appendix B.3.

**Implementation Details** All experiments are conducted on a single NVIDIA GeForce RTX 4090 GPU with 24 GB of memory, using Python 3.8 and PyTorch 2.1. We employ L2 regularization through the Adam optimizer (Kingma, 2014) with a weight decay of $5 \times 10^{-4}$ and $(\beta_1, \beta_2)$ set to $(0.9, 0.99)$, and use cross-entropy loss for optimization. All baseline models are trained for 100 epochs, and the initial learning rate is selected within $\{0.0001, 0.0005, 0.001, 0.005, 0.01\}$, and all hidden feature dimensions are tuned from the set $\{32, 64, 128, 256, 512, 1024\}$.

## 5.2 OVERALL PERFORMANCE COMPARISON

The overall comparison results are presented in Tables 1 and 2. Based on these results, we make several key observations. ❶ Among existing methods, DLinear performs the worst due to its simplistic architecture consisting only of linear layers, which limits its ability to capture informative temporal patterns. The model that combines GNN with contrastive learning, DGMSCL, achieves the second-best performance on HAR, ISRUC and PM2.5, likely because it leverages GCN to model inter-variable correlations, thereby extracting more informative features. On the Epilepsy dataset, PatchTST shows the best baseline performance, which can be attributed to its channel-independent learning strategy that benefits from better generalization under imbalanced conditions. Overall, TimeLT consistently achieves the highest performance across all datasets, demonstrating the strong

Table 2: Accuracy (ACC) results for head and tail classes. Specifically, the top 50% most frequent classes are designated as head classes, while the remaining classes are designated as tail classes.

| Methods | HAR-LT | | | | | | ISRUC-LT | | | | | | Epilepsy-LT | | | | | | PM2.5 | |
|---|---|---|---|---|---|---|---|---|---|---|---|---|---|---|---|---|---|---|---|---|
| | IR=10 | | IR=50 | | IR=100 | | IR=10 | | IR=50 | | IR=100 | | IR=10 | | IR=50 | | IR=100 | | IR=11.17 | |
| | Head | Tail | Head | Tail | Head | Tail | Head | Tail | Head | Tail | Head | Tail | Head | Tail | Head | Tail | Head | Tail | Head | Tail |
| DLinear | 85.72 | 24.43 | 86.46 | 10.96 | 87.23 | 7.90 | 35.75 | 11.59 | 35.78 | 10.37 | 34.46 | 9.25 | 50.00 | 0.00 | 50.00 | 0.00 | 50.00 | 0.00 | 92.06 | 28.36 |
| PatchTST | 77.86 | 90.30 | 79.88 | 77.91 | 72.37 | 74.99 | 64.90 | 56.86 | 80.70 | 34.62 | 78.40 | 27.90 | 73.01 | 52.22 | 70.14 | 35.75 | 66.91 | 26.59 | 80.40 | 32.48 |
| xPatch | 91.84 | 86.39 | 90.25 | 64.37 | 90.55 | 51.37 | 50.00 | 0.00 | 50.00 | 0.00 | 50.00 | 0.00 | 68.25 | 24.12 | 50.00 | 0.00 | 50.00 | 0.00 | 84.23 | 34.11 |
| TimeMixer++ | 87.34 | 93.88 | 86.99 | 58.78 | 86.26 | 52.17 | 75.91 | 36.82 | 74.67 | 13.80 | 69.22 | 6.86 | 50.00 | 0.00 | 50.00 | 0.00 | 50.00 | 0.00 | 84.30 | 33.26 |
| PatchMLP | 91.13 | 90.54 | 90.31 | 59.49 | 90.60 | 45.90 | 72.65 | 24.54 | 64.58 | 11.81 | 61.78 | 5.92 | 50.00 | 0.00 | 50.00 | 0.00 | 50.00 | 0.00 | 91.67 | 28.48 |
| MPTSNet | 89.60 | 91.49 | 87.23 | 74.21 | 86.14 | 65.70 | 75.42 | 49.03 | 69.30 | 32.45 | 66.18 | 23.14 | 69.05 | 35.60 | 66.54 | 29.39 | 61.34 | 24.30 | 87.52 | 27.63 |
| CFAMG | 91.70 | 88.32 | 87.97 | 60.95 | 87.23 | 50.31 | 47.34 | 18.60 | 45.13 | 12.49 | 46.24 | 8.00 | 38.70 | 10.44 | 50.00 | 0.00 | 50.00 | 0.00 | 86.31 | 31.29 |
| DGMSCL | 90.76 | 93.51 | 87.41 | 84.54 | 88.18 | 81.85 | 91.88 | 70.22 | 91.47 | 35.65 | 82.22 | 32.38 | 38.19 | 46.63 | 26.99 | 43.55 | 18.81 | 45.90 | 87.65 | 24.56 |
| TimeLT | 96.75 | 97.93 | 96.10 | 93.19 | 96.57 | 86.86 | 89.36 | 76.04 | 87.69 | 66.54 | 82.95 | 60.26 | 72.38 | 61.22 | 68.71 | 55.62 | 65.94 | 47.99 | 92.92 | 40.58 |

generalization ability. ❷ As the degree of class imbalance increases, most baseline models exhibit significant performance degradation, particularly on ISRUC and Epilepsy. When the imbalance ratio is high (e.g., IR≥50), several models (e.g., TimeMixer++ and PatchMLP) achieve an MCC score of zero, indicating that the models predict all samples as a single head class, failing to capture any class-wise discrimination. This highlights the severe challenge posed by extreme class imbalance in time series classification. ❸ We present fine-grained results for head and tail classes in Table 2. Most models perform adequately on head classes. However, they exhibit a significantly reduced accuracy, or even complete failure, when dealing with tail classes. In contrast, our model demonstrates superior performance on tail classes, with an accuracy improvement of up to 78.96%.

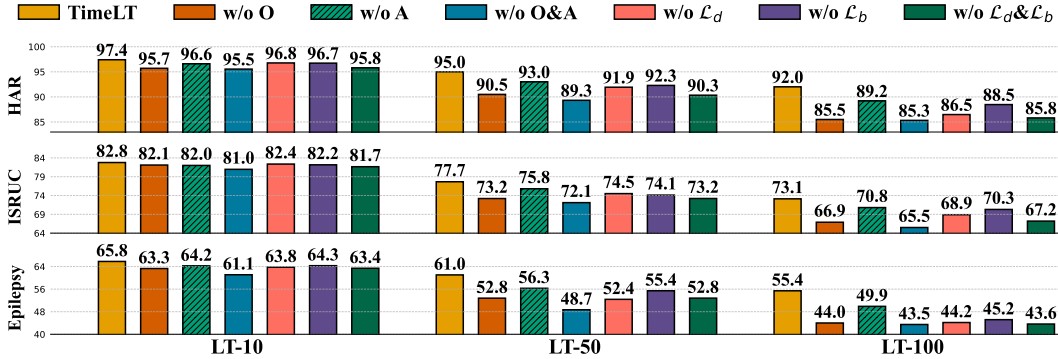

Figure 5: Ablation study evaluated with accuracy.

## 5.3 ABLATION STUDY

We conduct an ablation study to evaluate the effectiveness of each component of TimeLT, which includes four variants: 1) 'w/o O&A': The preprocessing approach (oversampling and data augmentation) is excluded. 2) 'w/o $\mathcal{L}_d$': Directional regularization is not considered in TimeLT. 3) 'w/o $\mathcal{L}_b$': Boundary-repelling regularization is not considered in TimeLT. 4) 'w/o $\mathcal{L}_d$&$\mathcal{L}_d$': Both boundary-repelling and directional regularization are not considered. These variants are compared with the full version of TimeLT.

Figure 5 presents the accuracy results across three datasets with imbalance ratios IR ∈ {10, 50, 100}. Overall, each module contributes positively to prediction performance. Specifically, removing oversampling and data augmentation results in a significant drop in accuracy (i.e., **w/o O&A**), particularly at higher imbalance ratios. This is because the preprocessing module enables the encoder to learn rich temporal representations for each class by applying varied degrees and quantities of augmentation. Moreover, in scenarios of extreme class imbalance (e.g., IR = 100), oversampling is generally more crucial than data augmentation. This is because when tail-class samples are extremely scarce, relying solely on augmentation may generate distorted samples based on limited data. Additionally, when the loss function $\mathcal{L}_d$ is removed, **w/o $\mathcal{L}_d$** only using boundary-repelling regularization

Table 3: Performance comparison with different downsampling methods.

| Methods | Metrics | HAR-LT | | | Epilepsy-LT | | |
|---|---|---|---|---|---|---|---|
| | | IR=10 | IR=50 | IR=100 | IR=10 | IR=50 | IR=100 |
| Average Pooling | ACC(%) | 97.41 | 94.99 | 92.01 | 65.81 | 61.04 | 55.41 |
| | F1(%) | 97.38 | 94.83 | 91.60 | 64.84 | 60.84 | 54.46 |
| | MCC | 0.969 | 0.940 | 0.905 | 0.582 | 0.520 | 0.453 |
| Max Pooling | ACC(%) | 97.07 | 94.51 | 91.63 | 62.43 | 56.04 | 48.50 |
| | F1(%) | 96.93 | 94.34 | 91.34 | 60.80 | 55.22 | 44.87 |
| | MCC | 0.962 | 0.934 | 0.902 | 0.545 | 0.471 | 0.380 |
| CNN | ACC(%) | 96.82 | 93.90 | 87.67 | 39.86 | 26.82 | 28.94 |
| | F1(%) | 96.80 | 93.65 | 87.13 | 33.11 | 20.77 | 22.22 |
| | MCC | 0.964 | 0.928 | 0.855 | 0.295 | 0.093 | 0.129 |

Table 4: Performance comparison of different backbones for global temporal modeling.

| Methods | Metrics | HAR-LT | | | ISRUC-LT | | |
|---|---|---|---|---|---|---|---|
| | | IR=10 | IR=50 | IR=100 | IR=10 | IR=50 | IR=100 |
| GRU | ACC(%) | 97.41 | 94.99 | 92.01 | 82.77 | 77.71 | 73.12 |
| | F1(%) | 97.38 | 94.83 | 91.60 | 80.29 | 75.77 | 70.28 |
| | MCC | 0.969 | 0.940 | 0.905 | 0.780 | 0.714 | 0.655 |
| MLP | ACC(%) | 90.16 | 78.41 | 75.32 | 65.48 | 56.16 | 51.33 |
| | F1(%) | 89.88 | 77.40 | 73.63 | 65.37 | 51.04 | 39.32 |
| | MCC | 0.883 | 0.745 | 0.712 | 0.560 | 0.442 | 0.385 |
| Transformer | ACC(%) | 88.91 | 85.66 | 81.18 | – | – | – |
| | F1(%) | 88.72 | 85.16 | 80.53 | – | – | – |
| | MCC | 0.867 | 0.830 | 0.775 | – | – | – |
| PatchTST | ACC(%) | 93.89 | 88.97 | 87.25 | 73.91 | 69.80 | 64.65 |
| | F1(%) | 93.20 | 87.60 | 86.48 | 72.90 | 67.50 | 61.92 |
| | MCC | 0.919 | 0.857 | 0.863 | 0.709 | 0.623 | 0.567 |
| iTransformer | ACC(%) | 90.86 | 87.46 | 83.03 | 66.64 | 62.83 | 57.22 |
| | F1(%) | 90.28 | 87.18 | 81.29 | 64.28 | 59.03 | 53.71 |
| | MCC | 0.890 | 0.862 | 0.797 | 0.609 | 0.573 | 0.484 |

alone increases the risk of samples being pushed into the feature spaces of other classes, leading to misclassification. Furthermore, although the variant **w/o** $\mathcal{L}_b$ achieves the second-best performance on the ISRUC and Epilepsy datasets, it still falls short of the complete TimeLT. This is primarily due to the neglect of distribution similarity across different categories in time series data, resulting in feature representations mapped closer to class boundaries. In summary, all components of TimeLT contribute effectively.

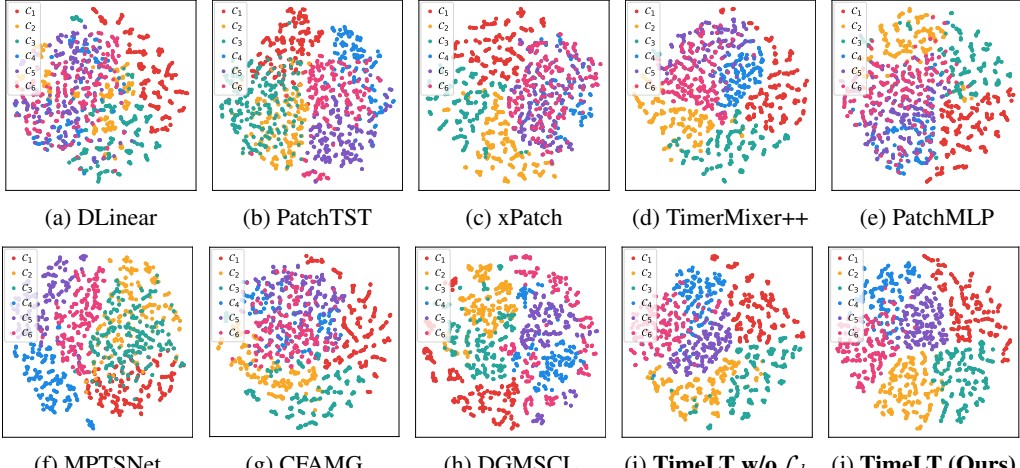

(a) DLinear  (b) PatchTST  (c) xPatch  (d) TimerMixer++  (e) PatchMLP

(f) MPTSNet  (g) CFAMG  (h) DGMSCL  (i) **TimeLT w/o** $\mathcal{L}_b$  (j) **TimeLT (Ours)**

Figure 6: Visualization of output representations on the HAR-LT-100 dataset.

## 5.4 VISUALIZATION ANALYSIS

To intuitively evaluate the quality of output representations learned by different time series models (i.e., $\mathbf{z}_n$ in Eq.(7)), we use t-SNE to visualize them on the HAT-LT-100 dataset. We compare TimeLT with representative models, particularly the variant 'w/o $\mathcal{L}_b$', which omits the boundary-repelling term. As shown in Figure 6, we can observe that the complete TimeLT generates more compact clusters for samples within the same class and provides clearer boundaries between samples of different classes. This can be attributed to the careful design of TimeLT, particularly its adversarial decision boundary regularization term.

## 5.5 ARCHITECTURAL SENSITIVITY ANALYSIS

**Downsampling Methods** To evaluate the effectiveness of different downsampling methods, we employ three commonly used approaches: average pooling, max pooling, and the conventional convolution operation. As shown in Table 3, average pooling consistently outperforms both max pooling and convolution operation. This is likely because average pooling preserves the overall trend through window-wise averaging, which enhances robustness to noise and extreme values. In contrast, max pooling retains only the maximum value, which can amplify noise or outliers. The

convolution operation may capture latent high-level features, but it does not intuitively reflect the multi-scale characteristics of time series, such as trends.

**Global Temporal Modeling** We compare representative backbones, including MLPs, GRUs, and the attention mechanisms used in Transformer-based models, as shown in Table 4. "-" indicates that the model ran out of memory due to the long sequence length of ISRUC. The RNN-based GRU outperforms the simple linear MLP and Transformers. Specifically, the MLP cannot capture high-level temporal features, resulting in much lower performance. It is noted that the recently popular Transformer-based methods still falls short of GRU. While Transformer attention captures pair-wise dependencies, it lacks the explicit sequential modeling capability of GRU.

## 5.6 SENSITIVITY ANALYSIS

**Analysis of $\beta$** We select $\beta$ values within $\{0, 0.2, 0.4, 0.6, 0.8, 1\}$, where $\beta = 0$ denotes no oversampling and $\beta = 1$ represents complete class balancing. As shown in Figure 6-(a), TimeLT achieves the optimal performance on the HAR, ISRUC, and Epilepsy datasets, when $\beta$ is set to 0.4 or 0.6. When $\beta$ exceeds this optimal range, the model experiences performance degradation, likely due to overfitting on tail classes, which compromises the model's ability to learn robust temporal representations.

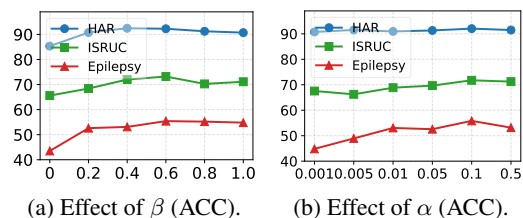

(a) Effect of $\beta$ (ACC).  (b) Effect of $\alpha$ (ACC).

Figure 7: Hyperparameter analysis on HAR-LT, ISRUC-LT, and Epilepsy-LT, with IR $= 100$.

**Analysis of $\alpha$** $\alpha$ controls the strength of perturbation added during data augmentation, increasing linearly from head classes to tail classes. We consider $\alpha$ values in $\{0.001, 0.005, 0.01, 0.05, 0.1, 0.5\}$. As shown in Figure 6-(b), we find that TimeLT with relatively larger $\alpha$ yield better performance, i.e., $\alpha = 0.1$. Both excessively small and large values of $\alpha$ are detrimental as small values lead to overfitting while large values induce substantial deviations of the augmented samples from the true data distribution.

**Efficiency Analysis** As shown in Figure 5, we compare the efficiency of TimeLT with PatchTST, TimeMixer++, and DGMSCL on two datasets with different temporal scales: HAR-LT-10 ($T = 128$) and ISRUC-LT-10 ($T = 3000$), where $T$ denotes input sequence length. PatchTST and DGMSCL are the second-best baselines. On HAR-LT-10, TimeLT uses 2.72 MB memory, 244.51 s training, and 0.0016 s per-sample inference. On ISRUC-LT-10, 2.90 MB, 704.87 s, and 0.01884 s, respectively. TimeLT achieves the highest accuracy while remaining lightweight and fast, showing that careful architectural design can boost performance without extra computational cost.

Table 5: Efficiency comparison on HAR-LT-10 and ISRUC-LT-10.

|  | Model | ACC(%) | Params(MB) | Train(s) | Inference(s) |
|---|---|---|---|---|---|
| HAR | PatchTST | 83.95 | 4.22 | 367.61 | 0.0042 |
|  | TimeMixer++ | 90.52 | 9.05 | 614.38 | 0.0277 |
|  | DGMSCL | 92.20 | 3.14 | 274.20 | 0.0031 |
|  | TimeLT | 97.41 | 2.72 | 244.51 | 0.0016 |
| ISRUC | PatchTST | 63.32 | 14.79 | 1843.24 | 0.0402 |
|  | TimeMixer++ | 56.77 | 155.84 | 2106.56 | 0.1320 |
|  | DGMSCL | 82.22 | 3.14 | 1354.90 | 0.0348 |
|  | TimeLT | 82.77 | 2.90 | 704.87 | 0.0188 |

## 6 CONCLUSION

In this paper, we propose TimeLT, a framework tailored for long-tail time series classification. It integrates a perturbation-aware data augmentation strategy to boost sample diversity, especially for underrepresented classes, a multi-scale temporal encoder for enhanced feature representation, and a boundary-repelling regularization term that fosters a well-separated and compact feature space. We also introduce a new long-tail TSC benchmark encompassing 16 models and multiple datasets, enabling fair comparisons and advancing future research. Experimental results demonstrate the effectiveness of TimeLT in long-tail TSC scenarios.

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

# A    RELATED WORK

## A.1    LONG-TAIL LEARNING

Long-tailed data distributions pose a persistent challenge in machine learning (Zhang et al., 2023; Yang et al., 2022), where class imbalance significantly degrades performance on tail classes. Existing approaches to this problem are typically grouped into three categories: (1) data re-sampling and augmentation, (2) specialized neural architectures, and (3) customized loss functions for embedding learning. The first approach balances class distributions through over-sampling, under-sampling, or synthetic data generation (Li et al., 2021; Shi et al., 2023). The second focuses on designing model structures that better capture discriminative features, especially for tail classes (Ahn et al., 2023; Wang et al., 2024b). The third line of work adapts loss functions, such as assigning higher weights to tail classes, to promote unbiased decision boundaries (Du et al., 2023; Wei & Gan, 2023). Time series classification models often inherit the long-tailed class distribution issue prevalent in vision and language tasks. However, this problem remains underexplored in TSC.

## A.2    MULTI-SCALE ANALYSIS IN TIME SERIES

Multi-scale analysis has demonstrated remarkable effectiveness in enhancing representation learning across various domains, including computer vision and multi-modal learning (Wang et al., 2021; Hu et al., 2020). Motivated by these successes, recent studies have extended this paradigm to the time series domain, which can be broadly categorized into patch-based and downsampling-based approaches. Patch-based methods divide the series into patches of varying lengths to model dependencies at different resolutions (Chang et al., 2025a). For example, Pathformer captures multi-scale embeddings using dual attention across patches (Chen et al., 2024), and LLM4TS employs a two-level aggregation on patches to embed multi-scale temporal information into a pre-trained LLM (Chang et al., 2025b). Downsampling-based methods generate temporally coarsened series via convolution or pooling, capturing dependencies at each scale (Liu et al., 2022). For instance, MAGNN constructs a dynamic multi-scale adaptive graph for modeling inter-variable relationships (Chen et al., 2023), while TimeMixer++ generates multi-scale representations to capture both short- and long-term patterns (Wang et al., 2024c). Compared with patch-based methods, downsampling-based approaches are better suited for capturing coarser-grained trend information.

# B    IMPLEMENTATION DETAILS

We summarized details of datasets, evaluation metrics, baseline models in this section.

## B.1    DATASETS

To verify the effectiveness of our TimeLT, we conduct experiments on four time series classification datasets: HAR[1] (Anguita et al., 2013), ISRUC-S3[2] (Khalighi et al., 2016), Epilepsy[3] (Andrzejak et al., 2001) and PM2.5[4]. The key statistical properties of all datasets are summarized in Table 6. Moreover, to evaluate model performance under various levels of class imbalance, we create three versions of the first three datasets with imbalance ratios (IR) set to IR$\in \{10, 50, 100\}$. The top 50% most frequent classes are designated as head classes, while the remaining classes are designated as tail classes.

Table 6: Dataset statistics.

| Dataset | HAR | ISRUC | Epilepsy | PM2.5 |
|---|---|---|---|---|
| #Train | 7,352 | 6,013 | 7,360 | 419,610 |
| #Val | 984 | 858 | 1,840 | 139,870 |
| #Test | 1,963 | 1,718 | 2,300 | 139,880 |
| Length | 128 | 3,000 | 178 | 168 |
| #Variable | 9 | 10 | 1 | 1 |
| #Class | 6 | 5 | 5 | 3 |

---

[1] https://paperswithcode.com/dataset/har

[2] https://sleeptight.isr.uc.pt/

[3] https://www.kaggle.com/datasets/harunshimanto/epileptic-seizure-recognition

[4] https://www.cnemc.cn/

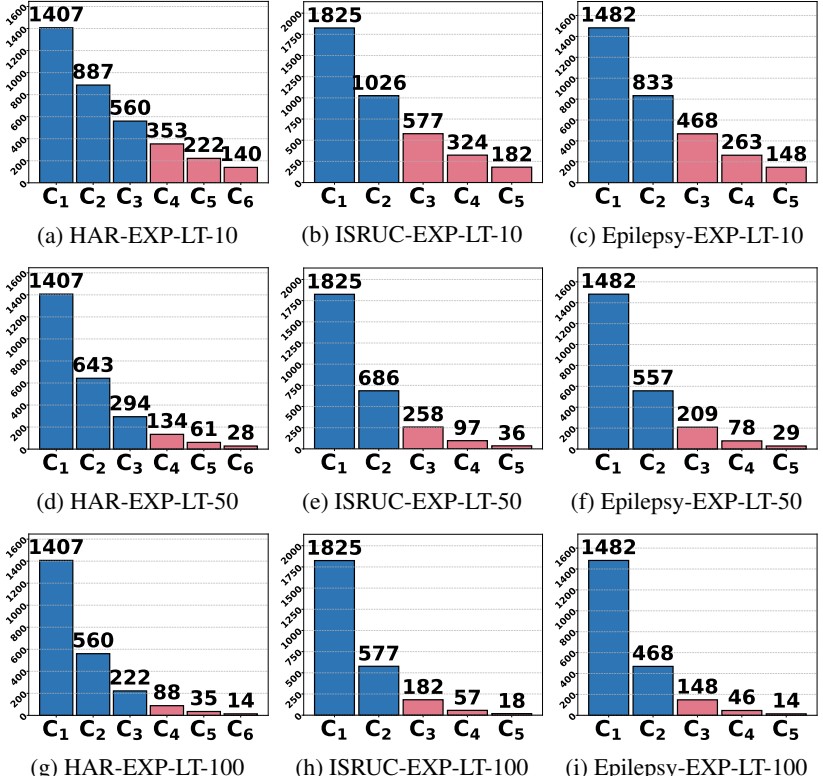

Figure 8: Exponential long-tailed distribution of HAR-LT, ISRUC-LT, and Epilepsy-LT datasets under three IR settings (10, 50, 100). The first three blue bars represent head classes and the last three red bars represent tail classes.

Class imbalance is generated through stratified random sampling without replacement from the original training set, resulting in an exponentially decaying class distribution (Wang et al., 2024e). The validation and test sets maintain their original distribution. The PM2.5 dataset is obtained from the real-time air quality monitoring platform maintained by the China National Environmental Monitoring Center, which provides hourly pollutant measurements from both urban and meteorological monitoring stations across China. We select 10 representative monitoring stations, each containing more than 70,000 hourly measurements collected over eight years, from January 1, 2016, to December 31, 2023. Using this dataset, PM2.5 readings from the past 168 time steps (i.e., one week) are used to predict pollution levels for the next day, categorized into three severity levels. This dataset exhibits a natural long-tail class distribution, with imbalance ratios of 11.17, 13.88, and 23.33 for the training, validation, and test sets, respectively. Figure 9 illustrates the class distributions of the three constructed long-tailed datasets under different imbalance ratios.

Existing datasets often have small sample sizes, which hinders the construction of realistic long-tailed distributions with higher imbalance ratios. They also exhibit similar imbalance ratios, reflecting limited diversity. In contrast, the datasets selected in this study are representative and capture diverse long-tailed scenarios. Notably, the PM2.5 dataset is sourced from real-world measurements rather than synthetically generated data.

## B.2 EVALUATION METRICS

We employ accuracy (ACC), F1-score (F1), and Matthews correlation coefficient (MCC) to assess performance. The calculations of these metrics are as follows:

Table 7: Full comparison results. **Bold** & Underline indicate the best & second best results, respectively. * denotes that the improvement over the second-best results is statistically significant according to a t-test with $p < 0.01$. IR represents the imbalance ratio. The HAR-LT, ISRUC-LT, and Epilepsy-LT datasets follow a long-tailed (exponential) distribution.

| Methods | Metrics | HAR-LT | | | ISRUC-LT | | | Epilepsy-LT | | | PM2.5 |
|---|---|---|---|---|---|---|---|---|---|---|---|
| | | IR=10 | IR=50 | IR=100 | IR=10 | IR=50 | IR=100 | IR=10 | IR=50 | IR=100 | IR=11.17 |
| Autoformer [NIPS 2021] | ACC(%) | 38.31 | 35.30 | 26.69 | 34.58 | 31.32 | 28.75 | 20.39 | 20.39 | 20.39 | 74.63 |
| | F1(%) | 27.79 | 28.05 | 19.82 | 31.17 | 22.36 | 20.53 | 6.77 | 6.77 | 6.77 | 47.04 |
| | MCC | 0.304 | 0.256 | 0.152 | 0.137 | 0.083 | 0.040 | 0.000 | 0.000 | 0.000 | 0.462 |
| Informer [AAAI 2021] | ACC(%) | 85.85 | 70.67 | 70.47 | 57.76 | 50.32 | 47.38 | 21.13 | 20.39 | 20.39 | 75.49 |
| | F1(%) | 85.55 | 69.32 | 68.30 | 52.13 | 38.47 | 33.54 | 8.13 | 6.77 | 6.77 | 48.43 |
| | MCC | 0.831 | 0.652 | 0.659 | 0.455 | 0.362 | 0.317 | 0.039 | 0.000 | 0.000 | 0.445 |
| Fedformer [ICML 2022] | ACC(%) | 62.89 | 33.57 | 21.44 | 44.41 | 37.66 | 34.16 | 40.13 | 25.82 | 20.39 | 77.28 |
| | F1(%) | 59.95 | 21.17 | 16.09 | 40.55 | 24.72 | 18.48 | 32.21 | 16.21 | 6.77 | 48.03 |
| | MCC | 0.567 | 0.254 | 0.073 | 0.279 | 0.160 | 0.091 | 0.293 | 0.122 | 0.000 | 0.463 |
| Paraformer [ICLR 2022] | ACC(%) | 77.23 | 68.47 | 61.53 | 45.32 | 37.79 | 32.50 | 45.09 | 42.04 | 20.39 | 78.46 |
| | F1(%) | 75.90 | 67.12 | 58.74 | 40.93 | 33.75 | 27.10 | 35.02 | 33.33 | 6.77 | 49.18 |
| | MCC | 0.734 | 0.628 | 0.543 | 0.405 | 0.323 | 0.278 | 0.389 | 0.330 | 0.000 | 0.428 |
| SegRNN [ArXiv 2023] | ACC(%) | 83.85 | 64.74 | 62.45 | 29.10 | 28.36 | 28.23 | 41.77 | 31.57 | 28.97 | 75.96 |
| | F1(%) | 83.58 | 62.83 | 57.57 | 22.92 | 18.49 | 16.69 | 33.43 | 23.25 | 20.58 | 47.70 |
| | MCC | 0.810 | 0.584 | 0.566 | 0.057 | 0.024 | 0.007 | 0.310 | 0.210 | 0.160 | 0.459 |
| DLinear [AAAI 2023] | ACC(%) | 56.75 | 50.84 | 49.61 | 25.43 | 24.83 | 23.72 | 20.39 | 20.39 | 20.39 | 78.42 |
| | F1(%) | 53.70 | 44.82 | 41.54 | 20.79 | 19.56 | 18.14 | 6.77 | 6.77 | 6.77 | 50.87 |
| | MCC | 0.500 | 0.467 | 0.458 | 0.019 | 0.006 | -0.008 | 0.000 | 0.000 | 0.000 | 0.474 |
| PatchTST [ICLR 2023] | ACC(%) | 83.95 | 75.85 | 74.02 | 63.32 | 59.08 | 53.14 | 60.62 | 49.92 | 43.14 | 73.37 |
| | F1(%) | 83.87 | 74.45 | 72.47 | 59.36 | 49.35 | 43.75 | 57.81 | 46.48 | 39.18 | 47.43 |
| | MCC | 0.809 | 0.717 | 0.699 | 0.544 | 0.488 | 0.400 | 0.532 | 0.417 | 0.347 | 0.410 |
| TimeXer [NIPS 2024] | ACC(%) | 89.04 | 74.52 | 73.10 | 66.82 | 60.82 | 56.28 | 20.39 | 20.39 | 20.39 | 78.62 |
| | F1(%) | 88.38 | 73.46 | 70.29 | 60.93 | 52.91 | 49.97 | 6.77 | 6.77 | 6.77 | 50.63 |
| | MCC | 0.868 | 0.698 | 0.688 | 0.575 | 0.499 | 0.443 | 0.000 | 0.000 | 0.000 | 0.505 |
| TimeMixer [ICLR 2024] | ACC(%) | 90.47 | 81.20 | 61.38 | 67.90 | 54.65 | 51.62 | 60.27 | 47.84 | 42.46 | 74.78 |
| | F1(%) | 90.51 | 80.92 | 58.12 | 61.35 | 42.70 | 38.58 | 57.77 | 43.92 | 36.78 | 46.50 |
| | MCC | 0.886 | 0.777 | 0.544 | 0.591 | 0.423 | 0.380 | 0.522 | 0.392 | 0.336 | 0.404 |
| iTransformer [ICLR 2024] | ACC(%) | 91.34 | 80.43 | 75.29 | 31.19 | 30.84 | 30.84 | 20.39 | 20.39 | 20.39 | 78.68 |
| | F1(%) | 91.16 | 79.39 | 72.82 | 13.66 | 12.33 | 10.22 | 6.77 | 6.77 | 6.77 | 48.96 |
| | MCC | 0.896 | 0.769 | 0.712 | 0.023 | 0.004 | -0.012 | 0.000 | 0.000 | 0.000 | 0.485 |
| xPatch [AAAI 2025] | ACC(%) | 89.51 | 77.48 | 72.08 | 19.73 | 19.73 | 19.73 | 45.37 | 20.39 | 20.39 | 76.51 |
| | F1(%) | 89.20 | 76.49 | 69.85 | 6.59 | 6.59 | 6.59 | 37.83 | 6.77 | 6.77 | 50.11 |
| | MCC | 0.874 | 0.731 | 0.671 | 0.000 | 0.000 | 0.000 | 0.381 | 0.000 | 0.000 | 0.470 |
| Timemixer++ [ICLR 2025] | ACC(%) | 90.52 | 73.61 | 69.63 | 56.77 | 47.48 | 42.20 | 20.39 | 20.39 | 20.39 | 76.61 |
| | F1(%) | 90.37 | 71.52 | 65.45 | 46.61 | 31.21 | 24.19 | 6.77 | 6.77 | 6.77 | 49.33 |
| | MCC | 0.888 | 0.691 | 0.652 | 0.462 | 0.329 | 0.230 | 0.000 | 0.000 | 0.000 | 0.470 |
| PatchMLP [AAAI 2025] | ACC(%) | 90.93 | 75.54 | 68.98 | 51.57 | 41.75 | 38.66 | 20.39 | 20.39 | 20.39 | 78.57 |
| | F1(%) | 90.81 | 74.33 | 65.07 | 42.10 | 27.92 | 23.46 | 6.77 | 6.77 | 6.77 | 50.38 |
| | MCC | 0.891 | 0.711 | 0.643 | 0.373 | 0.246 | 0.184 | 0.000 | 0.000 | 0.000 | 0.483 |
| MPTSNet [AAAI 2025] | ACC(%) | 90.65 | 78.83 | 76.87 | 65.48 | 56.64 | 53.96 | 51.02 | 45.20 | 42.65 | 75.96 |
| | F1(%) | 90.51 | 77.18 | 74.40 | 59.70 | 46.66 | 39.66 | 45.16 | 36.17 | 34.22 | 50.37 |
| | MCC | 0.889 | 0.754 | 0.731 | 0.551 | 0.446 | 0.411 | 0.414 | 0.360 | 0.328 | 0.493 |
| CFAMG [KDD 2025] | ACC(%) | 90.12 | 74.83 | 72.13 | 34.28 | 32.60 | 30.33 | 21.91 | 20.39 | 20.39 | 78.43 |
| | F1(%) | 90.03 | 73.09 | 69.77 | 28.49 | 25.56 | 24.12 | 11.89 | 6.78 | 6.78 | 52.04 |
| | MCC | 0.882 | 0.705 | 0.677 | 0.125 | 0.095 | 0.062 | 0.030 | 0.000 | 0.000 | 0.497 |
| DGMSCL [NN 2025] | ACC(%) | 92.20 | 86.04 | 86.36 | 82.22 | 65.19 | 58.73 | 43.97 | 37.39 | 35.48 | 78.75 |
| | F1(%) | 92.06 | 85.56 | 85.92 | 78.08 | 50.68 | 42.15 | 35.80 | 29.70 | 27.42 | 52.63 |
| | MCC | 0.907 | 0.836 | 0.839 | 0.763 | 0.578 | 0.523 | 0.376 | 0.294 | 0.255 | 0.508 |
| TimeLT (Ours) | ACC(%) | **97.41***  | **94.99*** | **92.01*** | **82.77*** | **77.71*** | **73.12*** | **65.81*** | **61.04*** | **55.41*** | **83.24*** |
| | F1(%) | **97.38*** | **94.83*** | **91.60*** | **80.29*** | **75.77*** | **70.28*** | **64.84*** | **60.84*** | **54.46*** | **61.26*** |
| | MCC | **0.969*** | **0.940*** | **0.905*** | **0.780*** | **0.714*** | **0.655*** | **0.582*** | **0.520*** | **0.453*** | **0.584*** |

$$\text{ACC} = \frac{TP + TN}{TP + FP + FN + TN},$$

$$\text{F1} = \frac{2 \times P \times R}{P + R},$$

$$\text{MCC} = \frac{TP \times TN - FP \times FN}{\sqrt{(TP + FP)(TP + FN)(TN + FP)(TN + FN)}},$$

where $P$ and $R$ represent the precision and recall, respectively. $TP$, $FN$, $TN$, and $FP$ denote true positives, false negatives, true negatives, and false positives, respectively.

## B.3 BASELINE MODELS

We conduct comparisons with 16 widely recognized state-of-the-art models. Table 7 presents the complete performance comparison.

- **Autoformer** (Wu et al., 2021) It integrates progressive series decomposition with an Auto-Correlation mechanism to efficiently capture long-range dependencies, surpassing self-attention in accuracy and speed.

- **Informer** (Zhou et al., 2021) Informer introduces ProbSparse attention, attention distilling, and a generative decoder to enable efficient and accurate long-sequence time series forecasting with reduced complexity and faster inference.

- **Fedformer** (Zhou et al., 2022) It integrates seasonal-trend decomposition and frequency-domain enhancement into Transformers to capture global and local patterns efficiently, achieving superior accuracy with linear complexity.

- **Pyraformer** (Liu et al., 2022) It introduces a pyramidal attention module that captures multi-resolution temporal dependencies via inter-scale trees and intra-scale connections. It achieves high forecasting accuracy with constant signal path length and linear efficiency

- **SegRNN** (Lin et al., 2023) It introduces segment-wise iterations and parallel multi-step forecasting to reduce recurrent steps in RNNs, enabling efficient and accurate long-term forecasting.

- **DLinear** (Zeng et al., 2023) It challenges Transformer-based models by using series decomposition to separate input time series into trend and seasonal components, then applying distinct linear layers to each component and summing their outputs for final predictions.

- **PatchTST** (Nie et al., 2022) It divides time series into fixed length patches and captures pair-wise dependencies using a vanilla Transformer encoder. Additionally, it operates in a channel independent manner, applying the encoder separately to each time series channel.

- **TimeXer** (Wang et al., 2024h) It integrates exogenous variables into time series forecasting using patch-wise self-attention and variate-wise cross-attention, enabling effective fusion of external and internal information.

- **TimeMixer** (Wang et al., 2024d) It adopts a fully MLP-based architecture to decompose and integrate multi-scale temporal patterns, enabling simultaneous long-term and short-term analysis by effectively decoupling complex historical information.

- **iTransformer** (Liu et al., 2023) It applies attention and feed-forward networks on variate tokens by embedding time points per variable, enhancing multivariate correlation learning. Without changing Transformer basics, it boosts forecasting accuracy and generalization for time series.

- **xPatch** (Stitsyuk & Choi, 2025) It introduces a dual-flow architecture characterized by channel-independent patching. It separates seasonal-trend components, enabling dual-stream prediction: MLP for trends and CNN for seasonal patterns.

- **Timemixer++** (Wang et al., 2024c) It transforms multi-scale series into multi-resolution time images and uses dual-axis attention with hierarchical mixing to capture complex temporal patterns.

- **PatchMLP** (Kong et al., 2025) It decomposes temporal features into trend and residual components using a moving average, and then enables channel-wise noise reduction as well as facilitates cross-variable interactions through channel mixing.

- **MPTSNet** (Mu et al., 2025) It decomposes time series into multi-periodic segments and learns both local patterns and global dependencies across scales. By combining convolution and attention, the model achieves stronger feature extraction and improved interpretability.

- **CFAMG** (Wang et al., 2025) It uses a disentangled classifier to identify causal factors distinguishing classes, then applies counterfactual intervention to generate synthetic minority samples that retain key structures while reflecting minority characteristics.

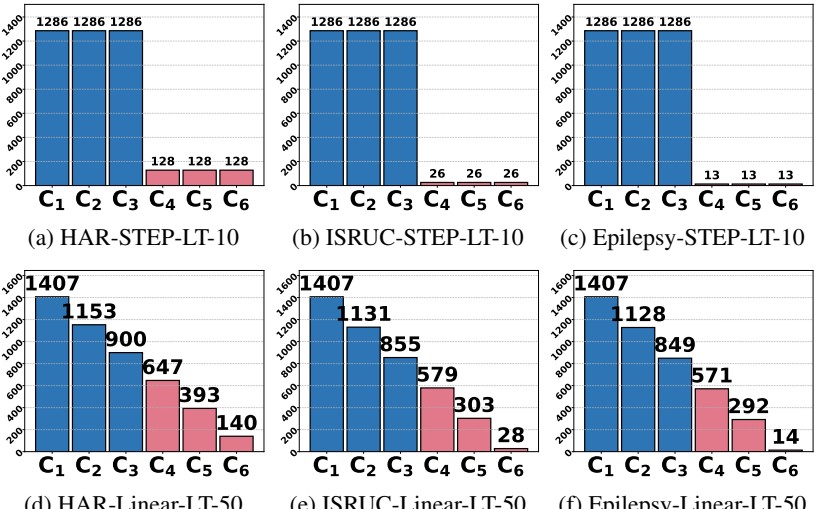

(a) HAR-STEP-LT-10  (b) ISRUC-STEP-LT-10  (c) Epilepsy-STEP-LT-10

(d) HAR-Linear-LT-50  (e) ISRUC-Linear-LT-50  (f) Epilepsy-Linear-LT-50

Figure 9: Step-wise and Linear long-tailed distribution of HAR-LT dataset under three IR settings (10, 50, 100).

Table 8: Overall performance and head/tail accuracy on the HAR-LT dataset under the **Step-wise** long-tailed distribution. Metrics include ACC(%), F1(%), MCC, Head(%), and Tail(%).

| Methods | IR=10 | | | | | IR=50 | | | | | IR=100 | | | | |
|---|---|---|---|---|---|---|---|---|---|---|---|---|---|---|---|
| | ACC | F1 | MCC | Head | Tail | ACC | F1 | MCC | Head | Tail | ACC | F1 | MCC | Head | Tail |
| DLinear | 49.01 | 41.19 | 0.459 | 86.34 | 7.78 | 50.68 | 42.59 | 0.474 | 88.84 | 9.67 | 48.39 | 37.69 | 0.436 | 90.17 | 3.52 |
| PatchTST | 79.97 | 79.50 | 0.760 | 80.64 | 78.55 | 73.96 | 72.94 | 0.689 | 80.72 | 65.51 | 65.43 | 62.95 | 0.591 | 81.61 | 46.41 |
| xPatch | 83.46 | 82.76 | 0.802 | 91.40 | 73.49 | 73.86 | 72.12 | 0.690 | 92.25 | 52.94 | 67.60 | 64.69 | 0.618 | 91.70 | 40.02 |
| TimeMixer++ | 88.20 | 87.91 | 0.859 | 92.52 | 83.41 | 71.93 | 68.00 | 0.684 | 87.81 | 54.00 | 66.07 | 62.46 | 0.603 | 89.76 | 38.38 |
| PatchMLP | 85.09 | 84.67 | 0.822 | 92.09 | 77.24 | 62.55 | 58.53 | 0.562 | 89.20 | 32.15 | 60.06 | 54.08 | 0.537 | 91.11 | 25.00 |
| MPTSNet | 90.98 | 90.83 | 0.892 | 90.67 | 91.04 | 75.55 | 74.42 | 0.708 | 90.02 | 55.62 | 67.19 | 64.97 | 0.612 | 88.61 | 42.44 |
| CFAMG | 83.65 | 83.31 | 0.804 | 89.72 | 76.67 | 69.43 | 66.17 | 0.639 | 89.91 | 46.11 | 57.16 | 51.62 | 0.509 | 90.01 | 20.74 |
| DGMSCL | 91.13 | 90.85 | 0.893 | 94.22 | 87.61 | 81.71 | 81.13 | 0.781 | 88.79 | 73.17 | 73.40 | 71.05 | 0.688 | 90.37 | 54.72 |
| **TimeLT (Ours)** | **96.06** | **96.01** | **0.952** | **96.41** | **95.59** | **89.64** | **89.07** | **0.878** | **92.84** | **85.49** | **82.61** | **81.52** | **0.793** | **94.96** | **68.09** |

- **DGMSCL** (Qian et al., 2025) It addresses imbalanced TSC via dynamic graph reconstruction and a mixed contrast loss. The loss combines weight-augmented inter-graph supervision and minority class-aware contrast to enhance representation.

## C HYPERPARAMETER ANALYSIS

**Analysis of $\lambda_1$ and $\lambda_2$**   $\lambda_1$ and $\lambda_2$ control the weights of $\mathcal{L}_b$ and $\mathcal{L}_d$, respectively. As shown in Figure 10, where optimal accuracy results are marked with stars, TimeLT achieves its optimal performance when both parameters reach 1. Smaller values result in insufficient repulsion from the decision boundary or repulsion in incorrect directions, while larger values diminish the effect of the classification loss.

**Analysis of $S$**   Figure 11 presents the experimental results with the hyperparameter $S$ selected from $\{1, 2, 3, 4, 5, 6\}$. We can observe that the optimal classification performance corresponding to different values of $S$ varies across datasets. Specifically, the best results are achieved with $S = 3$, 5, and 4 for the HAR, ISRUC, and Epilepsy datasets, respectively. When handling time series with longer time steps, more levels of downsampling are required. However, excessively large values of $S$ may lead to the extraction of overly coarse-grained features, which can confuse the model and consequently degrade predictive performance.

Table 9: Overall performance and head/tail accuracy on the HAR-LT dataset under the **Linear** long-tailed distribution. Metrics include ACC(%), F1(%), MCC, Head(%), and Tail(%).

| Methods | IR=10 | | | | | IR=50 | | | | | IR=100 | | | | |
|---|---|---|---|---|---|---|---|---|---|---|---|---|---|---|---|
| | ACC | F1 | MCC | Head | Tail | ACC | F1 | MCC | Head | Tail | ACC | F1 | MCC | Head | Tail |
| DLinear | 54.62 | 50.57 | 0.480 | 80.41 | 26.75 | 53.38 | 46.43 | 0.484 | 86.66 | 16.88 | 52.08 | 47.50 | 0.452 | 75.72 | 26.46 |
| PatchTST | 86.75 | 86.75 | 0.842 | 92.32 | 81.43 | 76.36 | 74.38 | 0.724 | 80.18 | 72.61 | 74.70 | 69.97 | 0.711 | 81.43 | 67.82 |
| xPatch | 89.14 | 88.86 | 0.870 | 91.47 | 86.21 | 82.90 | 82.30 | 0.795 | 88.05 | 76.35 | 82.42 | 81.57 | 0.791 | 90.83 | 72.77 |
| TimeMixer++ | 93.63 | 93.52 | 0.924 | 96.57 | 90.64 | 86.04 | 85.77 | 0.834 | 87.79 | 84.46 | 82.06 | 81.25 | 0.791 | 84.99 | 78.91 |
| PatchMLP | 91.21 | 91.00 | 0.895 | 93.03 | 89.34 | 85.99 | 85.60 | 0.833 | 89.34 | 82.27 | 78.45 | 75.25 | 0.753 | 90.15 | 66.10 |
| MPTSNet | 88.84 | 88.58 | 0.869 | 87.67 | 90.73 | 85.89 | 85.48 | 0.834 | 89.54 | 82.33 | 82.42 | 81.07 | 0.796 | 88.99 | 75.60 |
| CFAMG | 76.92 | 71.14 | 0.742 | 90.52 | 63.31 | 75.04 | 68.70 | 0.722 | 89.52 | 60.55 | 75.45 | 69.03 | 0.727 | 89.47 | 61.39 |
| DGMSCL | 90.23 | 90.08 | 0.884 | 88.13 | 92.15 | 81.59 | 79.52 | 0.788 | 88.61 | 74.35 | 78.19 | 71.86 | 0.758 | 91.45 | 64.70 |
| **TimeLT (Ours)** | **97.77** | **97.75** | **0.973** | **97.82** | **97.71** | **97.14** | **97.06** | **0.965** | **97.36** | **96.61** | **96.62** | **96.58** | **0.959** | **96.80** | **96.34** |

Table 10: Accuracy (%) of different augmentation methods under various imbalance ratios (IR).

| Methods | HAR-LT | | | ISRUC-LT | | | Epilepsy-LT | | |
|---|---|---|---|---|---|---|---|---|---|
| | IR=10 | IR=50 | IR=100 | IR=10 | IR=50 | IR=100 | IR=10 | IR=50 | IR=100 |
| FFT | 96.18 | 94.15 | 91.87 | 81.94 | 76.33 | 72.86 | 65.43 | 60.77 | 54.40 |
| Warping | 96.37 | 93.60 | 91.65 | 81.31 | 74.25 | 71.87 | 63.32 | 57.68 | 53.23 |
| Shuffle | 94.85 | 92.31 | 89.12 | 80.14 | 74.02 | 70.09 | 63.21 | 57.06 | 43.18 |
| **Personalized Augmentation** | **97.41** | **94.99** | **92.01** | **82.77** | **77.71** | **73.12** | **65.81** | **61.04** | **55.41** |

**Comparison with Alternative Augmentation Methods** We compare our personalized data augmentation with three typical methods: Fourier-based augmentation (FFT), Time warping (Warping) and Subsequence shuffling (Shuffling). *FFT* transforms the time series into the frequency domain, perturbs its frequency components, and reconstructs the variant series. *Warping* nonlinearly stretches or compresses the time axis. *Shuffling* divides the series into consecutive segments and randomly rearranges their order. As shown in Table 10, the experimental results indicate that in long-tail imbalanced scenarios, our personalized augmentation strategy achieves the best performance. In contrast, subsequence shuffling performs the worst, as it may disrupt the inherent sequential structure, making it difficult for multi-scale analysis to accurately capture trend-related features.

# D   DISCUSSION

While TimeLT demonstrates strong performance in long-tail TSC, it has certain limitations. Currently, our benchmark includes only a single real-world long-tailed dataset. In future work, we plan to collect more long-tailed time series datasets to further expand and diversify the benchmark, facilitating more comprehensive evaluation of long-tail TSC methods.

# E   ETHICS STATEMENT

Since our work is confined to the problem of time series forecasting, it poses no foreseeable ethical risks.

# F   REPRODUCIBILITY STATEMENT

The model architecture is strictly formalized with equations in the main text, along with all implementation details. To ensure reproducibility, the code will be made publicly available once the paper is accepted.

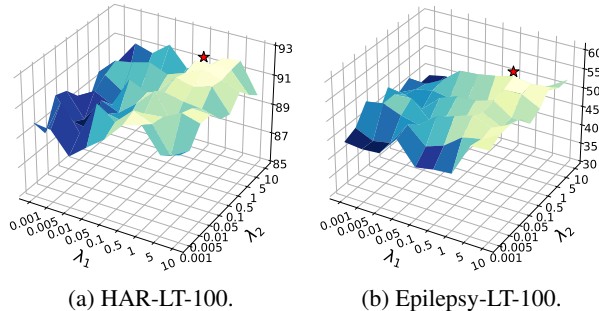

| (a) HAR-LT-100. | (b) Epilepsy-LT-100. |
| :---: | :---: |

Figure 10: Accuracy sensitivity of $\lambda_1$ and $\lambda_2$.

Figure 11: Effect of $S$ (ACC).

## G  LLM USAGE STATEMENT

We only use LLMs as a language optimization tool to polish sentences, improving their readability and fluency.

Table 11: Comparison with different downsampling methods.

| Methods | Metrics | HAR-LT | | | ISRUC-LT | | | Epilepsy-LT | | | PM2.5 |
| :---: | :---: | :---: | :---: | :---: | :---: | :---: | :---: | :---: | :---: | :---: | :---: |
| | | IR=10 | IR=50 | IR=100 | IR=10 | IR=50 | IR=100 | IR=10 | IR=50 | IR=100 | IR=11.17 |
| Average Pooling | ACC(%) | 97.41 | 94.99 | 92.01 | 82.77 | 77.71 | 73.12 | 65.81 | 61.04 | 55.41 | 83.24 |
| | F1(%) | 97.38 | 94.83 | 91.60 | 80.29 | 75.77 | 70.28 | 64.84 | 60.84 | 54.46 | 61.26 |
| | MCC | 0.969 | 0.940 | 0.905 | 0.780 | 0.714 | 0.655 | 0.582 | 0.520 | 0.453 | 0.584 |
| Max Pooling | ACC(%) | 97.07 | 94.51 | 91.63 | 59.82 | 74.10 | 62.01 | 62.43 | 56.04 | 48.50 | 78.00 |
| | F1(%) | 96.93 | 94.34 | 91.34 | 56.29 | 68.69 | 55.11 | 60.80 | 55.22 | 44.87 | 60.14 |
| | MCC | 0.962 | 0.934 | 0.902 | 0.553 | 0.674 | 0.546 | 0.545 | 0.471 | 0.380 | 0.490 |
| CNN | ACC(%) | 96.82 | 93.90 | 87.67 | 61.85 | 45.63 | 59.76 | 39.86 | 26.82 | 28.94 | 79.10 |
| | F1(%) | 96.80 | 93.65 | 87.13 | 58.97 | 41.93 | 55.58 | 33.11 | 20.77 | 22.22 | 54.21 |
| | MCC | 0.964 | 0.928 | 0.855 | 0.516 | 0.349 | 0.492 | 0.295 | 0.093 | 0.129 | 0.495 |

Table 12: Performance comparison of GRU and Transformer for global temporal modeling on four datasets. "-" indicates that the model ran out of memory due to the long sequence length of ISRUC.

| Methods | Metrics | HAR-LT | | | ISRUC-LT | | | Epilepsy-LT | | | PM2.5 |
| :---: | :---: | :---: | :---: | :---: | :---: | :---: | :---: | :---: | :---: | :---: | :---: |
| | | IR=10 | IR=50 | IR=100 | IR=10 | IR=50 | IR=100 | IR=10 | IR=50 | IR=100 | IR=11.17 |
| GRU | ACC(%) | 97.41 | 94.99 | 92.01 | 82.77 | 77.71 | 73.12 | 65.81 | 61.04 | 55.41 | 83.24 |
| | F1(%) | 97.38 | 94.83 | 91.60 | 80.29 | 75.77 | 70.28 | 64.84 | 60.84 | 54.46 | 61.26 |
| | MCC | 0.969 | 0.940 | 0.905 | 0.780 | 0.714 | 0.655 | 0.582 | 0.520 | 0.453 | 0.584 |
| MLP | ACC(%) | 90.16 | 78.41 | 75.32 | 65.48 | 56.16 | 51.33 | 59.26 | 50.63 | 40.41 | 79.56 |
| | F1(%) | 89.88 | 77.40 | 73.63 | 65.37 | 51.04 | 39.32 | 58.55 | 48.83 | 36.33 | 53.48 |
| | MCC | 0.883 | 0.745 | 0.712 | 0.560 | 0.442 | 0.385 | 0.493 | 0.408 | 0.294 | 0.521 |
| Transformer | ACC(%) | 88.91 | 85.66 | 81.18 | - | - | - | 44.79 | 44.20 | 39.97 | 79.15 |
| | F1(%) | 88.72 | 85.16 | 80.53 | - | - | - | 41.67 | 40.02 | 35.23 | 52.71 |
| | MCC | 0.867 | 0.830 | 0.775 | - | - | - | 0.339 | 0.339 | 0.278 | 0.503 |

