# OpenReview forum: "Toward Robust Feature Space in Long-Tailed Time Series Classification: A Multi-Scale Perspective"
_ICLR.cc/2026/Conference — Submitted to ICLR 2026_

### Official Review · Reviewer_PqPE · 2025-10-24

**Soundness:** 2
**Presentation:** 3
**Contribution:** 2
**Rating:** 2
**Confidence:** 4

**Summary:**

This paper proposes TimeLT to tackle long-tail time series classification by considering multi-scal temporal encoding, data augmentation, and variant representation learning strategies. TimeLT demonstrates effectiveness across 4 selected datasets with ablation study and parameter analysis.

**Strengths:**

1. The idea of boundary-repelling regularization $L_b$ sounds interesting and novel to me, which encourages samples to be away to other classes, with the corresponding ablation study and parameter study to support this claim.
2. The paper details the parameters and the training configurations, especially that the authors include the codebase as the supplementary, making this work reproducible.
3. The ablation study and parameter analysis are extensive and comprehensive, with testing variants of each module, hyper-parameters, and the loss designs.

**Weaknesses:**

1. The motivation about higher inter-class similarity for time series data is not convincing (Fig. 2 with L65-67). While dogs and pandas are different that can be easily separated, if we consider more fine-grained classes (e.g., black bear vs. brown bear), they probably may not be highly separable. Note that this concern is raised since the authors use "walking" and "running" as the time series examples, which is unfair to compare to coarse categories in the image domain. As this is the main (and only) motivation, this work does not sound necessary, even though it may demonstrate performance improvements.
2. While TimeLT demonstrates effectiveness over existing baselines, the experiments are only conducted with 4 datasets; however, CFAMG used 53 time-series datasets for evaluations, causing the experiments in this paper lacking insufficient validation. Additionally, the used 4 datasets are different from the CFAMG paper, which is difficult to evaluate its validity from the tables.
3. The third contribution of releasing a new benchmark remains unclear and questionable since the authors do not describe any motivations about it as well as why existing benchmarks are insufficient (e.g., UCR and UEA in the CFAMG paper)
4. The multi-scale temporal encoding has been explored by existing works, e.g., [1, 2]. However, the authors do not include any related works for comparisons and discussions.
5. The setting for head and tail classes (Table 2) is a bit confusing. Since it is an imbalance task, dividing with 50-50 to represent head and tail could largely cover classes that have similar portions. This raises the validity of interpreting Table 2.
6. [Minor]: Missing reference in L676.

[1] LLM4TS: Aligning Pre-Trained LLMs as Data-Efficient Time-Series Forecasters

[2] PromptTSS: A Prompting-Based Approach for Interactive Multi-Granularity Time Series Segmentation

**Questions:**

Q1: Regarding the claim "representations of tail classes tend to cluster near or even overlap with those of head classes, leading to blurred decision boundaries", could the authors prove or provide references to support this claim?
Q2: What is the reason that many methods suffer from very worse performance for Epilepsy-LT?

---

> ### Author Response · Authors · 2025-11-22
> **Part I of the Response**
>
> > **W1. The motivation about higher inter-class similarity for time series data is not convincing (Fig. 2 with L65-67). While dogs and pandas are different that can be easily separated, if we consider more fine-grained classes (e.g., black bear vs. brown bear), they probably may not be highly separable. Note that this concern is raised since the authors use "walking" and "running" as the time series examples, which is unfair to compare to coarse categories in the image domain. As this is the main (and only) motivation, this work does not sound necessary, even though it may demonstrate performance improvements.**
>
>
> We thank the reviewer for the valuable comments. The motivation of our work is grounded in empirical evidence obtained from multiple datasets, and the figures included in the paper are provided to corroborate the clarity of this motivation. To address your concern, we selected fine-grained image samples and quantitatively analyzed the JS divergence between fine-grained classes: “black bear” vs. “brown bear” yielded 0.303, and “white tiger” vs. “tiger” yielded 0.417, both higher than the corresponding values in our HAR-LT and Epilepsy-LT datasets (Figure 3 in the revised manuscript). This supports our motivation that high inter-class similarity in long-tailed time series presents a significant challenge.
>
> > **W2. While TimeLT demonstrates effectiveness over existing baselines, the experiments are only conducted with 4 datasets; however, CFAMG used 53 time-series datasets for evaluations, causing the experiments in this paper lacking insufficient validation. Additionally, the used 4 datasets are different from the CFAMG paper, which is difficult to evaluate its validity from the tables.**
>
> We thank the reviewers for their valuable feedback. We would like to clarify that our study is not limited to only four datasets. Each of the HAR, ISRUC, and Epilepsy datasets contains three distinct sub-datasets. The novelty of our dataset collection lies in the following **two aspects**:
>
> (1) We have additionally incorporated a real-world PM2.5 dataset derived from actual measurements. **This dataset is naturally imbalanced, allowing for a more realistic evaluation of the model's practical applicability**. **In contrast, the CFAMG datasets are manually generated.**
>
> (2) The datasets included in CFAMG are generally **small in scale**, making it difficult to construct more challenging scenarios, such as those with an Imbalance Ratio (IR) of 100. In fact, many of the 53 datasets in CFAMG are also **redundant**, with imbalance ratios ranging only between 2 and 5 (e.g., DodgerLoopDay, Earthquakes, ECG5000, EthanolLevel, FreezerRegularTrain, GunPoint, GunPointAgeSpan, Ham, HandMovementDirection, etc.). In comparison, our dataset collection includes **larger and more varied datasets**, enabling a more comprehensive evaluation of model effectiveness across varying degrees of imbalance.
>
> > **W3. The third contribution of releasing a new benchmark remains unclear and questionable since the authors do not describe any motivations about it as well as why existing benchmarks are insufficient (e.g., UCR and UEA in the CFAMG paper).**
>
> Thanks for the valuable comments. Please refer to W2.
>
> > **W4. The multi-scale temporal encoding has been explored by existing works, e.g., [1, 2]. However, the authors do not include any related works for comparisons and discussions.**
>
> Thanks for the valuable comment. We have added related work on multi-scale analysis. The content of the related work on multi-scale analysis is provided below, and the complete version with references is included in Appendix A.
>
> - **Added discussion.** Multi-scale analysis has demonstrated remarkable effectiveness in enhancing representation learning across various domains, including computer vision and multi-modal learning (Wang et al., 2021; Hu et al., 2020). Motivated by these successes, recent studies have extended this paradigm to the time series domain, which can be broadly categorized into patch-based and downsampling-based approaches. Patch-based methods divide the series into patches of varying lengths to model dependencies at different resolutions (Chang et al., 2025a). For example, Pathformer captures multi-scale embeddings using dual attention across patches (Chen et al., 2024), and LLM4TS employs a twolevel aggregation on patches to embed multi-scale temporal information into a pre-trained LLM (Chang et al., 2025b). Downsampling-based methods generate temporally coarsened series via convolution or pooling, capturing dependencies at each scale (Liu et al., 2022). For instance, MAGNN constructs a dynamic multi-scale adaptive graph for modeling inter-variable relationships (Chen et al., 2023), while TimeMixer++ generates multi-scale representations to capture both short- and long-term patterns (Wang et al., 2024c). Compared with patch-based methods, downsampling-based approaches are better suited for capturing coarser-grained trend information.

---

> ### Author Response · Authors · 2025-11-22
> **Part II of the Response**
>
> > **W5. The setting for head and tail classes (Table 2) is a bit confusing. Since it is an imbalance task, dividing with 50-50 to represent head and tail could largely cover classes that have similar portions. This raises the validity of interpreting Table 2.**
>
> We thank the reviewers for their valuable questions. In fact, the 50-50 split is a widely adopted strategy in recent long-tailed learning research [1, 2] designed to ensure comprehensive coverage of all classes during performance evaluation. This approach does not compromise the accuracy of performance reporting, as the sample size of head classes still substantially exceeds that of tail classes. To address concerns of this nature, we have included class-wise performance results for the ISRUC-LT-100 dataset in the table below as a concrete illustration.
>
> | Model           | Class 1 | Class 2 | Class 3 | Class 4 | Class 5 |
> |-----------------|----------|----------|----------|----------|----------|
> | DLinear         | 42.97    | 25.96    | 13.43    | 6.66     | 7.66     |
> | PatchTST     | 63.01    | 93.80    | 64.37    | 3.21     | 16.12    |
> | TimeMixer        | 78.32    | 76.03    | 48.13    | 2.54     | 11.47    |
> | iTransformer   | 96.84    | 2.20     | 0.00     | 0.00     | 0.00     |
> | xPatch       | 100      | 0        | 0        | 0        | 0        |
> | TimeMixer++     | 90.48    | 47.97    | 20.58    | 0.26     | 0.47     |
> | PatchMLP      | 90.81    | 32.76    | 17.05    |0.26     | 0.47     |
> | TimeLT       | 84.01    | 81.90    | 62.25    |  53.65    | 64.89    |
>
> [1] Long-Tailed Anomaly Detection with Learnable Class Names.
> [2] Long-Tailed Out-of-Distribution Detection: Prioritizing Attention to Tail.
>
> > **W6. [Minor]: Missing reference in L676.**
>
> We thank the reviewer for the comment. The missing reference at L676 has been added.
>
>
> > **Q1: Regarding the claim "representations of tail classes tend to cluster near or even overlap with those of head classes, leading to blurred decision boundaries", could the authors prove or provide references to support this claim?**
>
> We thank the reviewers for their valuable questions. The primary challenge in long-tailed learning is the model's tendency to underfit tail classes and misclassify them as head classes. This occurs because the feature space of tail classes often overlaps with or is even encompassed by that of head classes, leading to erroneous decision boundaries. This issue is further exacerbated in time-series data due to the high inter-class similarity. As demonstrated in Figure 6 of our paper, the feature spaces of classes generated by many advanced time-series analysis models indeed exhibit significant overlap.
>
> > **Q2: What is the reason that many methods suffer from very worse performance for Epilepsy-LT?**
>
> We thank the reviewer for the valuable question. We analyzed the inter-class similarity using the Jensen-Shannon divergence and found that the classes in the Epilepsy-LT dataset exhibit higher similarity, as shown in the Figure 3 of the revised manuscript. Despite this challenge, our method still achieves the best performance, which can be attributed to the design of our model.

---

> > ### Author Response · Authors · 2025-11-22
> > **Part III of the Response**
> >
> > We sincerely thank Reviewer #PqPE for the constructive comments. Your suggestions have contributed to improving the work. We sincerely appreciate your time and consideration.

---

> ### Comment · Reviewer_PqPE · 2025-11-27
>
> I appreciate the authors for their response and for improving the manuscript. I have raised my score to 4 since there remain some concerns.
>
> > W1
>
> Thanks for providing an additional analysis. While it may be more convincing to some extent, I would have room that this does support the motivation for this paper. We can see that multiple classes in CIFAR-LT still has minimal differences (e.g., C2 vs. C10; C3 vs. C10), indicating that they are not highly separable as claimed in the paper. In that case, if we could draw similar distribution figures for such image cases, I believe the figure would look similar to the time-series cases as shown in Figure 2. Therefore, the motivation for this paper remains unconvincing to me.
>
> > W2 & W3
>
> Thanks for the clarification, which is reasonable to me. It would be better to highlight this note somewhere in the manuscript.
>
> > W4
>
> While the authors provide discussions as the related work section, the authors do not compare them or clarify the difference between them, i.e., the discussion only introduces about their approaches.
>
> > Q2
>
> Could the authors elaborate exactly what design of the proposed model leads to the improvements on Epilepsy-LT? Although the authors provide potential root cause by analyzing the similarity, the reason why the proposed method not suffering from this issue remains unknown.

---

> > ### Author Response · Authors · 2025-12-01
> > **Further response to the reviewer's feedback**
> >
> > - **Further response to W1**
> >
> > We thank the reviewer for the valuable comments. As indicated by the JS divergence matrices, 64% of class pairs in CIFAR-LT exhibit a divergence greater than 0.1, whereas only 26% and 0% of class pairs in HAR and Epilepsy exceed this threshold. Thus, we find that inter-class similarity is generally higher in time-series datasets.
> >
> > - **Further response to W2\&W3**
> >
> > We thank the reviewer for the valuable comments. This dataset comparison is highlighted in Appendix B.1 of the manuscript.
> >
> > - **Further response to W4**
> >
> > We thank the reviewer for the valuable comments. Multi-scale analysis has demonstrated remarkable effectiveness in enhancing representation learning across various domains, including computer vision and multi-modal learning. Motivated by these successes, recent studies have extended this paradigm to the time series domain, which can be broadly categorized into patch-based and downsampling-based approaches. Patch-based methods divide the time series into patches of varying lengths to model dependencies at different resolutions. For example, Pathformer captures multi-scale embeddings using dual attention across patches, and LLM4TS employs a two-level aggregation on patches to embed multi-scale temporal information into a pre-trained LLM. Downsampling-based methods generate temporally coarsened series via convolution or pooling, capturing dependencies at each scale. MAGNN uses CNNs to extract multi-scale features from time series and then constructs a dynamic multi-scale adaptive graph to model inter-variable relationships, while TimeMixer++ employs CNNs to generate multi-scale representations capturing both short- and long-term patterns. TimeMixer uses average pooling to generate multi-scale time series and then decouples each scale into trend and seasonal components for prediction.
> >
> > **Compared with patch-based methods, which capture dependencies between sub-sequences of the same scale and across sub-sequences of different scales, downsampling-based approaches are more effective at extracting coarser-grained trend information. Although CNN-based downsampling methods can achieve good performance by implicitly capturing multi-scale features, this implicitness may confuse subsequent modules. In contrast, average pooling explicitly preserves the trend information of the time series. Although multi-scale analysis contributes to the enrichment of tail-class features, it is insufficient on its own to fully address the challenges posed by the issue of long-tailed TSC. Furthermore, we also conduct comprehensive experiments to compare and analyze the effects of different downsampling methods on long-tail classification, as shown in Table 3 of the manuscript.**
> >
> > In addition to comparing multi-scale analysis methods such as Pyraformer, TimeMixer, and TimeMixer++, we have also included a patch-based approach, i.e., Pathformer. The results are shown in the following Table.
> >
> > | Method      | Metric | HAR IR=10 | HAR IR=50 | HAR IR=100 | ISRUC IR=10 | ISRUC IR=50 | ISRUC IR=100 |
> > |------------|--------|-----------|-----------|------------|-------------|-------------|--------------|
> > | Pathformer | ACC(%) | 88.89     | 81.25     | 75.75      | 56.98       | 54.30       | 45.59        |
> > | Pathformer | F1(%)  | 88.76     | 80.46     | 74.49      | 44.56       | 35.39       | 26.19        |
> > | Pathformer | MCC(%) | 0.867     | 0.776     | 0.715      | 0.436       | 0.400       | 0.291        |
> > | TimeLT (Ours)| ACC(%) | 97.41     | 94.99     | 92.01      | 82.77       | 77.71       | 73.12        |
> > | TimeLT (Ours)| F1(%)  | 97.38     | 94.83     | 91.60      | 80.29       | 75.77       | 70.28        |
> > | TimeLT (Ours)| MCC    | 0.969     | 0.940     | 0.905      | 0.780       | 0.714       | 0.655        |
> >
> > - **Further response to Q2**
> >
> > We thank the reviewer for the valuable comments. In long-tailed scenarios, models tend to be biased toward head classes due to their dominant sample sizes. In time-series data, the high inter-class similarity further exacerbates this issue, making tail-class features more likely to overlap with those of head classes and thereby reducing discriminative ability. Our ablation study demonstrates that the primary improvements on Epilepsy-LT stem from two key components: the personalized augmentation strategy and the boundary-repelling regularization. The personalized augmentation effectively increases the diversity of tail samples, mitigating the strong bias toward head classes. More importantly, the boundary-repelling term encourages similar features to contract toward their respective class centers while moving away from inter-class decision boundaries, resulting in a more discriminative feature space. This is crucial for addressing the class overlap caused by high inter-class similarity, and explains why our method still achieves optimal performance on this dataset.

---

### Official Review · Reviewer_5fq8 · 2025-10-30

**Soundness:** 2
**Presentation:** 2
**Contribution:** 3
**Rating:** 4
**Confidence:** 3

**Summary:**

This paper proposes TimeLT, a method designed to address the challenges of imbalanced long-tailed time series classification (TSC).
TimeLT is composed of three main steps. Firstly, data augmentation via oversampling and perturbation is employed to enhance data diversity. Next, a temporal encoder is implemented for time series representation, consisting of a 1D-CNN embedding on multi-scale down-sampled series and a GRU as the backbone encoder. Finally, two regularization terms are applied to refine the decision boundary, with one pushing samples away from the boundary and the other pulling embeddings closer to the corresponding centroid.
Experimental results across multiple datasets, in comparison with 16 baseline methods, demonstrate TimeLT's leading performance on long-tailed TSC. These results are further verified through analysis including ablation, visualization, and sensitivity test.

**Strengths:**

(1) The problem of long-tailed time series classification is important and timely within the community. The proposed method provides a well-motivated solution. \
(2) The experimental results are compelling and demonstrate the efficacy of the method. \
(3) The structure of the article is clear, and the paper is easy to follow.

**Weaknesses:**

(1) The ablation is not rigorous. Specifically, in the preprocessing stage, oversampling is a commonplace method when dealing with imbalanced classes. The current "**w/o O&A**" ablation does not clarify whether the improvements stem from oversampling or perturbation. A more thorough ablation should include "**w/o O**", "**w/o A**", and "**w/o O&A**" to ascertain the contribution of each component. \
(2) The novelty of the proposed framework appears to be more of an engineering integration rather than an algorithmic breakthrough, given its reliance on established techniques. \
(3) There are several problems w.r.t. the writing. For example, grammatical faults like "This is can be" and format faults like "**Analysis of** $\beta$" ($\beta$ should be bolded) \
(4) Please refer to the questions for further potential weaknesses.

**Questions:**

(1) The Imbalance Ratio (IR) is defined as the ratio between the largest and smallest class sample counts. This only considers the two extreme classes. For instance, compare two scenarios: one where class sizes decay linearly from $N_1$ to $N_C$, and another where only $N_C$ drops sharply with the others being similar. Under your IR definition and its application in the method, would there be different effects in these cases? \
(2)  In Eq.(3), the use of c/C presumes a linear decay of sample counts across classes. Is this assumption valid? Or would it be more appropriate to use $N_c$ / $N_C$? \
(3) The author suggests GRU is a better choice in comparison with the Transformer because of the sequential modelling capability. Two questions have arisen w.r.t. this argument. Firstly, there are certain variations of the Transformer models that can model temporal dependencies better than the vanilla Transformer, but these variations are not included in the analysis or the comparable experiments. Second, the author suggests that information leakage caused by the Transformer will cause less accurate modelling. However, variations like causal attention enable the model only having access to the past time steps. At the same time, the whole series is available for TSC, which differentiates it from time series forecasting, so I do not consider that there is an information leakage problem. \
(4) Can Eq.(8) and Eq.(9) be unified in one contrastive loss function that simultaneously pushes away from the boundary (negative sample pair) and pulls closer to the centroid (positive sample pair)? \
(5) The Related Work in Appendix A has only one subsection A.1, and only the long-tail learning is included. Is this part left to be unfinished?

---

> ### Author Response · Authors · 2025-11-22
> **Part I of the Response**
>
> > **W1. The ablation is not rigorous. Specifically, in the preprocessing stage, oversampling is a commonplace method when dealing with imbalanced classes. The current "w/o O&A" ablation does not clarify whether the improvements stem from oversampling or perturbation. A more thorough ablation should include "w/o O", "w/o A", and "w/o O&A" to ascertain the contribution of each component.**
>
> We thank the reviewer for the valuable comment. We initially treated the personalized augmentation as an integrated module and therefore only reported the w/o O&A results. We agree with the reviewer that a more fine-grained ablation is necessary. To more clearly isolate the contribution of each component, we have added the w/o O and w/o A ablation experiments, as shown in the following Table. As the imbalance ratio increases, oversampling is more important, since augmentation alone may produce distorted samples when the available tail-class data are too limited.
>
> | Dataset  | IR  | Full | w/o A | w/o O | w/o O&A |
> |----------|-----|------|-------|-------|----------|
> | HAR      | 10  | 97.4 | 96.6  | 95.7  | 95.5     |
> | HAR      | 50  | 95.0 | 93.0  | 90.5  | 89.3     |
> | HAR      | 100 | 92.0 | 89.2  | 85.5  | 85.3     |
> | ISRUC    | 10  | 82.8 | 82.0  | 82.1  | 81.0     |
> | ISRUC    | 50  | 77.7 | 75.8  | 73.2  | 72.1     |
> | ISRUC    | 100 | 73.1 | 70.8  | 66.9  | 65.6     |
> | Epilepsy | 10  | 65.8 | 64.2  | 63.3  | 61.1     |
> | Epilepsy | 50  | 61.0 | 56.3  | 52.8  | 48.7     |
> | Epilepsy | 100 | 55.4 | 49.9  | 44.0  | 43.5     |
>
> > **W2. The novelty of the proposed framework appears to be more of an engineering integration rather than an algorithmic breakthrough, given its reliance on established techniques.**
>
>
> We thank the reviewer for the insightful comment.
> The contribution includes two perspectives.
>
> **(1) Problem level.** We reveal that long-tailed time-series data exhibit higher inter-class similarity than images, leading to overlapping temporal features and severely degraded tail-class separability, which has been overlooked in prior work and aligns with the conference’s scope of application-driven problem, as also reflected in several recent studies [1,2,3].  Moreover, existing time series studies often overlook practical applications and have largely reached saturation in model design.
>
> **(2) Method level.** Our method builds upon existing techniques while introducing improvements beyond a mere engineering integration. Specifically, first, existing methods typically treat all classes equally, whereas our personalized augmentation introduces a two-phase, class-adaptive oversampling strategy that first expands tail classes through inverse imbalance sampling and then applies progressively stronger, class-dependent perturbations to effectively enhance data diversity and robustness for scarce classes. Moreover, for the regularization strategy, we propose a novel boundary-repelling regularization, which adaptively estimates class decision boundaries to better enhance the separability of tail classes.
>
> **(3)  Benchmark.** we release a new long-tailed time series classification benchmark. Compared with existing benchmarks, our setup supports more data-sampling protocols, better reflecting real-world long-tailed distributions. Existing benchmarks’ long-tailed datasets are all synthetically generated; by contrast, we also include a real-world long-tailed time-series dataset that collects PM2.5 air-quality levels from 1,314 stations across mainland China. Because pollution events are rare, this dataset is naturally imbalanced, enabling evaluation of models in realistic scenarios.
>
> Reference：
>
> [1] Out-of-distribution Representation Learning for Time Series Classification. ICLR 2023.
>
> [2] S4M: S4 for multivariate time series forecasting with Missing values. ICLR 2025.
>
> [3] BrainUICL: An Unsupervised Individual Continual Learning Framework for EEG Applications. ICLR 2025.
>
> > **W3. There are several problems w.r.t. the writing. For example, grammatical faults like "This is can be" and format faults like "Analysis of $\beta$" ($\beta$ should be bolded)**
>
> We appreciate the reviewer’s careful reading and helpful comments. We have thoroughly reviewed the entire manuscript to correct grammatical and formatting errors. These issues have been corrected in the revised version.

---

> > ### Comment · Reviewer_5fq8 · 2025-11-25
> > **Further question towards the response of W2.**
> >
> > Specifically for (2) Method level, the oversampling and perturbation are both straightforward solutions when it comes to imbalanced data modelling. Though I consider the insight from the problem ( long-tailed time-series data exhibit higher inter-class similarity than images) is illustrative, I still cannot convince myself to consider the proposed methodology a novelty.

---

> ### Author Response · Authors · 2025-11-22
> **Part II of the Response**
>
> >  **Q1. The Imbalance Ratio (IR) is defined as the ratio between the largest and smallest class sample counts. This only considers the two extreme classes. For instance, compare two scenarios: one where class sizes decay linearly from $N_1$ to $N_C$, and another where only $N_C$ drops sharply with the others being similar. Under your IR definition and its application in the method, would there be different effects in these cases?**
>
> We thank the reviewer for the insightful question. To address the concern, we conduct experiments on the HAR-LT dataset using both Linear and Step-wise long-tailed distributions, as illustrated in Figure 9 of the revised manuscript.
>
> The corresponding results are summarized in the following table. We observe that our method consistently outperforms all existing baselines under both distribution settings. For comprehensive results, please refer to Tables 8 and 9 in the revised manuscript.
>
> | Stepped Distribution(IR=100) | ACC   | F1    | MCC   | Head  | Tail  |
> |------------------------------|-------|-------|-------|-------|-------|
> | DLinear                      | 48.39 | 37.69 | 0.436 | 90.17 | 3.52  |
> | PatchTST                     | 65.43 | 62.95 | 0.591 | 81.61 | 46.41 |
> | xPatch                       | 67.60 | 64.69 | 0.618 | 91.70 | 40.02 |
> | TimeMixer++                  | 66.07 | 62.46 | 0.603 | 89.76 | 38.38 |
> | PatchMLP                     | 60.06 | 54.08 | 0.537 | 91.11 | 25.00 |
> | MPTSNet                      | 67.19 | 64.97 | 0.612 | 88.61 | 42.44 |
> | CFAMG                        | 57.16 | 51.62 | 0.509 | 90.01 | 20.74 |
> | DGMSCL                       | 73.40 | 71.05 | 0.688 | 90.37 | 54.72 |
> | **TimeLT (Ours)**            | **82.61** | **81.52** | **0.793** | **94.96** | **68.09** |
>
> | Linear Distribution(IR=100)| ACC   | F1    | MCC   | Head  | Tail  |
> |-------------------|-------|-------|-------|-------|-------|
> | DLinear           | 52.08 | 47.50 | 0.452 | 75.72 | 26.46 |
> | PatchTST          | 74.70 | 69.97 | 0.711 | 81.43 | 67.82 |
> | xPatch            | 82.42 | 81.57 | 0.791 | 90.83 | 72.77 |
> | TimeMixer++       | 82.06 | 81.25 | 0.791 | 84.99 | 78.91 |
> | PatchMLP          | 78.45 | 75.25 | 0.753 | 90.15 | 66.10 |
> | MPTSNet           | 82.42 | 81.07 | 0.796 | 88.99 | 75.60 |
> | CFAMG             | 75.45 | 69.03 | 0.727 | 89.47 | 61.39 |
> | DGMSCL            | 78.19 | 71.86 | 0.758 | 91.45 | 64.70 |
> | **TimeLT (Ours)** | **96.62** | **96.58** | **0.959** | **96.80** | **96.34** |
>
> In fact, in addition to the artificially synthesized datasets, we also introduced a naturally imbalanced time series dataset in our paper to demonstrate effectiveness in real-world scenarios.
>
> >  **Q2. In Eq.(3), the use of c/C presumes a linear decay of sample counts across classes. Is this assumption valid? Or would it be more appropriate to use $N_c/N_C$.**
>
> Thank you very much for your insightful comment. In fact, this was a writing error; our original intention was to customize the perturbation based on the number of samples in each class. Therefore, this should be corrected to: $c$ → $N_C-N_{c}$ and $C$ → $N_C$. Sorry for the confusion.

---

> > ### Comment · Reviewer_5fq8 · 2025-11-25
> > **Further question towards the response of Q1**
> >
> > Thanks for your further experiments. The results seems  that TimeLT handles both Linear and Step-wise imbalance better than the baselines.
> > Still, I want to know if the definition of IR is reasonable, theoretically. Since the definition involve only the value from the two end but not depict the imbalance in the middle.

---

> > > ### Author Response · Authors · 2025-11-26
> > > **Response to the follow-up question regarding Q1**
> > >
> > > We thank the reviewer for the insightful feedback. The IR of our work follows the standard definition used in previous studies [1,2,3], which employs the ratio between the most frequent and least frequent classes to characterize overall imbalance and has become a widely adopted paradigm for defining long-tailed problems. Since the data distribution can take various forms, such as linear, exponential, or stepwise, it is not feasible to define IR based on the intermediate classes. Theoretically, the disparity in sample numbers between the two extreme classes is the primary driver of long-tailed bias. Models naturally tend to favor head classes with abundant samples, and this bias is largely induced by the extreme imbalance between the endpoints. Thus, we construct datasets with varying IR values of 10, 50, and 100. In addition, as shown in Figures 8,9, the intermediate classes in the datasets are also imbalanced.
> > >
> > > [1] A Systematic Review on Long-Tailed Learning.
> > >
> > > [2] Long-Tailed Anomaly Detection with Learnable Class Names.
> > >
> > > [3] Long-Tailed Out-of-Distribution Detection: Prioritizing Attention to Tail.

---

> ### Author Response · Authors · 2025-11-22
> **Part III of the Response**
>
> > **Q3. The author suggests GRU is a better choice in comparison with the Transformer because of the sequential modelling capability. Two questions have arisen w.r.t. this argument. Firstly, there are certain variations of the Transformer models that can model temporal dependencies better than the vanilla Transformer, but these variations are not included in the analysis or the comparable experiments. Second, the author suggests that information leakage caused by the Transformer will cause less accurate modelling. However, variations like causal attention enable the model only having access to the past time steps. At the same time, the whole series is available for TSC, which differentiates it from time series forecasting, so I do not consider that there is an information leakage problem.**
>
> We thank the reviewer for the valuable question. To address this concern, we divide the question into two sub-questions.
>
> **Q3.1 Variations of the Transformer models**
>
> To address the reviewer’s concern, we additionally evaluate two strong Transformer-based variants, PatchTST (subsequence-level modeling) and iTransformer (variable-level modeling). As shown in the following Table, their accuracies on HAR-LT and ISRUC-LT remain consistently lower than GRU. For example, on HAR-LT-10, PatchTST and iTransformer achieve 93.89% and 90.86% ACC, respectively, compared to 97.41% from our GRU-based design. Similar performance gaps are observed on ISRUC-LT.
>
> | Dataset  | IR  | GRU   | PatchTST | iTransformer |
> |----------|-----|-------|----------|--------------|
> | HAR-LT   | 10  | 97.41 | 93.89    | 90.86        |
> | HAR-LT   | 50  | 94.99 | 88.97    | 87.46        |
> | HAR-LT   | 100 | 92.01 | 87.25    | 83.03        |
> | ISRUC-LT | 10  | 82.77 | 73.91    | 66.64        |
> | ISRUC-LT | 50  | 77.71 | 69.80    | 62.83        |
> | ISRUC-LT | 100 | 73.12 | 64.65    | 57.22        |
>
> **Q3.2 Information Leakage of Transformer**
>
> We apologize for any confusion caused. Our intended meaning was that, since Transformers lack strong inductive bias, the attention coefficients computed between arbitrary time steps are unordered. As a result, they cannot explicitly model the unidirectional dependency from past to future time steps. Based on your suggestion, we have revised the ambiguous text as follows:
> Although recent Transformer-based methods capture pairwise dependencies through attention mechanisms, their weak inductive bias with respect to temporal order results in poorer performance for modeling sequential dependencies compared to GRU. Moreover, in imbalanced scenarios, the complexity of such architectures can lead to overfitting to the data.
>
> > **Q4. Can Eq.(8) and Eq.(9) be unified in one contrastive loss function that simultaneously pushes away from the boundary (negative sample pair) and pulls closer to the centroid (positive sample pair)?**
>
> We thank the reviewers for their valuable questions. Please allow us to clarify that although our approach shares a similar idea with contrastive loss, it is achieved by comparing cluster centroid representations, which differs from traditional contrastive learning methods based on sample pairs.
>
> Morover, while the two objectives can be mathematically merged into a single contrastive loss, in complex dataset scenarios, we may need to adjust the weights of the two losses to enable flexible learning. Moreover, keeping the two losses separate also allows for an intuitive presentation of their individual contributions.
>
> > **Q5. The Related Work in Appendix A has only one subsection A.1, and only the long-tail learning is included. Is this part left to be unfinished?**
>
> We apologize for the inadvertent omission that caused confusion. Essentially, this section serves as a supplement to the related work, which is why we used the \subsection command. Following the other reviewers' suggestion, we have now incorporated a discussion on multi-scale analysis into this section, thereby addressing this issue.

---

> > ### Author Response · Authors · 2025-11-22
> > **Part IV of the Response**
> >
> > We would like to thank Reviewer #5fq8 for the insightful observations and helpful recommendations. Your comments guided us to improve our article. We appreciate your thorough and considerate review.

---

> > ### Comment · Reviewer_5fq8 · 2025-11-25
> > **Further question towards the response of Q3.2.**
> >
> > Since the task here is time series classification, the whole segment of time series is available. Why the unidirectional modelling is still problem? I thought we can use both direction of time flow, unless you are doing the online classification where time series is gained in a real time flow.

---

> ### Author Response · Authors · 2025-11-26
> **Response to the follow-up question regarding Q3.2.**
>
> We thank the reviewer for the valuable feedback. In fact, Reference [1] suggests that introducing a directional attention mechanism can better capture temporal dynamics, and we have adopted this perspective to analyze potential reasons for the suboptimal performance of the Transformer. To avoid further confusion, we have revised our analysis accordingly. However, the main objective of this paper is to address the long-tail time series classification problem, rather than to overcome the limitations inherent to the Transformer architecture. We also appreciate the reviewer’s suggestion regarding the impact of forward and backward temporal modeling on long-tail time series classification. This is indeed an interesting research direction, and we plan to explore it in our future work.
>
> [1] TimeFormer: Transformer with Attention Modulation Empowered by Temporal Characteristics for Time Series Forecasting.

---

> ### Author Response · Authors · 2025-11-26
> **Response to the follow-up question regarding W2**
>
> We thank the reviewer for the insightful feedback. We summarize and rank our contributions as follows.
>
> (1) The problem-level insight: we reveal the high inter-class similarity and severe tail-class overlap in long-tailed time-series data.
>
> (2) A newly proposed boundary-repelling regularization that adaptively estimates class decision boundaries and pushes samples away from them, thereby enhancing tail-class separability.
>
> (3) A new long-tailed time-series benchmark that includes multiple long-tailed datasets, including a real-world dataset, as well as a library of various long-tailed TSC models. Our method achieves state-of-the-art performance.
>
> (4) Personalized augmentation and oversampling.
>
> (5) A multi-scale analysis approach.
>
> Overall, our primary contributions lie in the first three aspects. Although personalized augmentation is an simple extension of existing techniques, it addresses one aspect of the proposed problem and still contributes effectively to the overall performance improvement.
>
> We hope the reviewer could reconsider the novelty of our work in light of these contributions, particularly the first three.

---

### Official Review · Reviewer_Xejq · 2025-10-31

**Soundness:** 2
**Presentation:** 3
**Contribution:** 3
**Rating:** 6
**Confidence:** 2

**Summary:**

This paper propose a framework, named TimeLT, to learn a robust feature space from long-tailed data, thereby improving the overall accuracy for long-tailed TSC. Additionally, a benchmark is released, which includes data processing protocols, diverse datasets, and multiple baselines.

**Strengths:**

1.	The problem addressed in this paper is common and significant within the field of time-series classification.
2.	The authors publish a standardized benchmark that holds significant practical value for promoting fair comparisons and future development within the long-tail TSC domain.
3.	The experiments are comprehensive, demonstrating the proposed method's effectiveness through comparisons with multiple baselines.

**Weaknesses:**

1.	The components of TimeLT, including perturbation-aware data augmentation, multi-scale temporal encoding, and oundary-repulsion regularization, are well-established techniques that have been extensively studied, which to some extent limits the novelty of this paper. Authors should provide a more detailed explanation of why this combination can address the challenges posed by high inter-class similarity.
2.	This paper lacks discussion of the computational overhead and training/inference time of the TimeLT framework, as multi-scale encoding will inevitably increase computational costs.
3.	The description of the decision boundary embedding set B in Eq.(7) is vague, a more detailed computational process and explanation should be provided.

**Questions:**

The authors only illustrate inter-class similarity through Figure 2, lacking quantitative analysis and description (such as DTW distance) to demonstrate that it is indeed higher than typical image datasets.

---

> ### Author Response · Authors · 2025-11-22
> **Response to Reviewer Xejq**
>
> > **W1. The components of TimeLT, including perturbation-aware data augmentation, multi-scale temporal encoding, and boundary-repulsion regularization, are well-established techniques that have been extensively studied, which to some extent limits the novelty of this paper. Authors should provide a more detailed explanation of why this combination can address the challenges posed by high inter-class similarity.**
>
> We thank the reviewer for the insightful comment. Our design directly addresses the challenges we face. The specific challenges and corresponding design choices are as follows:
> - C1 Tail classes have few samples and require data augmentation. However, conventional oversampling or reweighting applies uniform augmentation across classes, which can easily lead to overfitting.
> - C2 When the model’s representation capacity is limited, similar data distributions can result in overlapping feature distributions.
> - C3 Overlapping feature distributions hinder the model from establishing well-separated decision boundaries.
>
> The proposed solutions are:
>
> - To C1 Personalized augmentation: We propose a novel augmentation strategy for scarce tail samples, preventing overfitting unlike oversampling or uniform augmentation.
> - To C2 Multi-scale modeling: Multi-scale analysis enables the model to capture more informative patterns, such as trend variations, thereby enhancing its ability to learn features of tail classes.
> - To C3 Boundary-Repelling regularization: Encourages the model to learn more discriminative decision boundaries, effectively separating head and tail classes.
>
>
> > **W2. This paper lacks discussion of the computational overhead and training/inference time of the TimeLT framework, as multi-scale encoding will inevitably increase computational costs.**
>
> We thank the reviewer for the insightful comment. As shown in the following Table, we compare the efficiency of TimeLT with several representative models, PatchTST, TimeMixer++, and DGMSCL, on two datasets with different temporal scales: HAR-LT-10 with a short lookback window ($T = 128$) and ISRUC-LT-10 with a long lookback window ($T = 3000$), where $T$ denotes the input sequence length. Among these baselines, PatchTST and DGMSCL serve as the second-best performers. For the HAR-LT-10 dataset, TimeLT requires 2.72 MB of memory, 244.51s of training time, and 0.0016s for single-sample inference. For the ISRUC-LT-10 dataset, the corresponding values are 2.90 MB, 704.87s, and 0.01884s, respectively. TimeLT achieves the highest accuracy while maintaining a lightweight model design and fast inference, which demonstrates that careful architectural design can improve performance without increasing computational cost.
>
> | Dataset | Model        | ACC (%) | Params (MB) | Train (s) | Inference (s) |
> |---------|-------------|---------|-------------|-----------|---------------|
> | HAR     | PatchTST    | 83.95   | 4.22        | 367.61    | 0.0042        |
> | HAR     | TimeMixer++ | 90.52   | 9.05        | 614.38    | 0.0277        |
> | HAR     | DGMSCL      | 92.20   | 3.14        | 274.20    | 0.0031        |
> | HAR     | TimeLT      | 97.41   | 2.72        | 244.51    | 0.0016        |
> | ISRUC   | PatchTST    | 63.32   | 14.79       | 1843.24   | 0.0402        |
> | ISRUC   | TimeMixer++ | 56.77   | 155.84      | 2106.56   | 0.1320        |
> | ISRUC   | DGMSCL      | 82.22   | 3.14        | 1354.90   | 0.0348        |
> | ISRUC   | TimeLT      | 82.77   | 2.90        | 704.87    | 0.0188        |
>
> > **W3. The description of the decision boundary embedding set B in Eq.(7) is vague, a more detailed computational process and explanation should be provided.**
>
> We thank the reviewer for the insightful comment. For class $\mathcal{C}\_i$, the set of decision boundary embeddings $\mathcal{B}\_i$ is defined as: $\mathcal{B}\_i = \\{ \mathbf{b}\_{i,1}, \mathbf{b}\_{i,2}, \dots, \mathbf{b}\_{i,C} \\} \setminus \mathbf{b}\_{i,i}$, which includes the decision boundary embeddings between class $i$ and all other classes $j \neq i$.
>
> > **Q1. The authors only illustrate inter-class similarity through Figure 2, lacking quantitative analysis and description (such as DTW distance) to demonstrate that it is indeed higher than typical image datasets.**
>
> We thank the reviewer for the insightful question. To further substantiate, we computed the JS divergence between class-wise distributions on the canonical image dataset CIFAR-LT and our HAR-LT and Epilepsy datasets. Due to limitations of the OpenReview rebuttal system, images cannot be displayed directly. You may refer to the revised Figure 3 for details. CIFAR-LT exhibits larger divergence values, indicating clearer inter-class separability. In contrast, HAR-LT and Epilepsy display smaller divergence values, highlighting the higher inter-class similarity inherent in time series data.

---

> > ### Author Response · Authors · 2025-11-22
> > **Response to Reviewer Xejq**
> >
> > We are grateful to Reviewer #Xejq for the detailed and thoughtful feedback. Your suggestions greatly helped us enhance the clarity and completeness of the paper. We truly appreciate your valuable review.

---

> > > ### Comment · Reviewer_Xejq · 2025-11-28
> > >
> > > I appreciate the authors' response, which has solved most of my concerns. However, W2 remains unanswered at the theoretical level. Could the authors provide a clear theoretical explanation as to why TimeLT attains higher computational efficiency compared with the other methods?

---

> > > > ### Author Response · Authors · 2025-12-01
> > > > **Response to computational efficiency compared with the other methods**
> > > >
> > > > We thank the reviewer’s valuable comments and perform a theoretical analysis of the time complexity.
> > > >
> > > > - PatchTST:
> > > >   $ N \times (T \times D^2 + P^2 \times D + P \times D^2) $
> > > >
> > > > - DGMSCL:
> > > >   $ 3N \times (T \times N \times D + T \times D^2 + T^2 \times D + T \times D^2) $
> > > >
> > > > - TimeMixer++:
> > > >   $ M \times \left( T \log T + \frac{T}{P} \times P \times D^2 + \frac{T}{P} \times D^2 + \left( \frac{T}{P} \right)^2 \times D + P \times D^2 + P^2 \times D + 2P \times \frac{T}{P} \times D^2 \right) $
> > > >
> > > > - Ours:
> > > > $ M \times N \times (T \times D^2 + T \times D^2) $
> > > >
> > > > Here, $T$ represents the time step, $N$ is the number of variables, $P$ is the number of subsequences, and $D$ is the maximum feature dimension. We observe that as the time dimension increases, the time complexity of other methods increases at a higher rate due to terms like $T^2$ or $P^2$. In contrast, our method has a time complexity that increases linearly with $T$, making it more efficient as the time dimension grows. This indicates that our method scales better with larger time steps, resulting in a lower increase in complexity compared to other approaches.

---

### Official Review · Reviewer_sotr · 2025-11-01

**Soundness:** 3
**Presentation:** 3
**Contribution:** 3
**Rating:** 6
**Confidence:** 4

**Summary:**

The paper tackles the problem of long-tailed time series classification, where models often fail under severe class imbalance. The authors propose TimeLT, which integrates (1) a perturbation-aware augmentation to enhance tail-class diversity, (2) a multi-scale temporal encoder for rich feature extraction, and (3) a boundary-repulsion regularization to improve class separability. A new benchmark with 16 baselines is introduced, and experiments show consistent gains, especially on tail classes.

**Strengths:**

1. The paper tackles an underexplored yet practically relevant problem, long-tailed time series classification, and provides a unified benchmark that may serve as a foundation for future research.

2. Extensive experiments across 16 baselines and multiple datasets convincingly show the effectiveness and robustness of TimeLT.

**Weaknesses:**

1. The augmentation and regularization strategies appear to reuse existing ideas with limited novelty or justification specific to time-series data.

2. The motivation-to-method alignment is weak and some design choices are introduced abruptly without sufficient reasoning.

**Questions:**

1. The abstract lists three components of TimeLT, but the unsolved problems they connected to are unclear.

2.  How does Gaussian noise outperform more structured augmentations such as frequency- or context-based methods?

3. What role does the multi-scale, channel-independent design play in handling class imbalance?

4.  The boundary-repulsion loss resembles margin-based or supervised contrastive objectives. Could the authors clarify its unique contribution and explain how it improves discrimination in long-tailed TSC?

---

> ### Author Response · Authors · 2025-11-22
> **Part I of the Response**
>
> **Thank you very much for your time and effort. Your feedback has been immensely helpful in improving the quality of our paper. In the following, we will address each of your suggestions point by point.**
>
> > **W1. The augmentation and regularization strategies appear to reuse existing ideas with limited novelty or justification specific to time-series data.**
>
> We thank the reviewer for the insightful comment. The contribution includes two perspectives.
>
> **(1) Problem level.** We reveal that long-tailed time-series data exhibit higher inter-class similarity than images, leading to overlapping temporal features and severely degraded tail-class separability, which has been overlooked in prior work and aligns with the conference’s scope of application-driven problem, as also reflected in several recent studies [1,2,3]. Moreover, existing time series studies often overlook practical applications and have largely reached saturation in model design.
>
> **(2) Method level.** Our approach is not a simple reuse of existing methods, but is specifically designed to address the unique challenges in long-tailed time series learning. First, existing methods typically treat all classes equally, whereas our personalized augmentation introduces a two-phase, class-adaptive oversampling strategy that first expands tail classes through inverse imbalance sampling and then applies progressively stronger, class-dependent perturbations to effectively enhance data diversity and robustness for scarce classes. Moreover, for the regularization strategy, we propose a novel boundary-repelling regularization, which adaptively estimates class decision boundaries to better enhance the separability of tail classes.
>
> Reference：
>
> [1] Out-of-distribution Representation Learning for Time Series Classification. ICLR 2023.
>
> [2] S4M: S4 for multivariate time series forecasting with Missing values. ICLR 2025.
>
> [3] BrainUICL: An Unsupervised Individual Continual Learning Framework for EEG Applications. ICLR 2025.
>
> >**W2. The motivation-to-method alignment is weak and some design choices are introduced abruptly without sufficient reasoning.**
>
> We thank the reviewer for the insightful comment. The motivation of our work is that data-level inter-class similarity in long-tailed time series leads the model to learn similar features and makes tail classes harder to distinguish.
> The challenges are as follows:
>
> - C1. Tail classes have few samples, and relying solely on oversampling or uniform data augmentation can easily lead to overfitting.
> - C2. When the model’s representation capacity is limited, similar data distributions can result in overlapping feature distributions.
> - C3. Overlapping feature distributions hinder the model from establishing well-separated decision boundaries.
>
> The corresponding design are as follows：
>
> - To C1. Personalized augmentation: We propose a novel augmentation strategy for scarce tail samples, preventing overfitting unlike oversampling or uniform augmentation.
> - To C2. Multi-scale modeling: Multi-scale analysis enables the model to capture more informative patterns, such as trend variations, thereby enhancing its ability to learn features of tail classes.
> - To C3. Boundary-Repelling regularization: Encourages the model to learn more discriminative decision boundaries, effectively separating head and tail classes.
>
> > **Q1. The abstract lists three components of TimeLT, but the unsolved problems they connected to are unclear.**
>
> We thank the reviewer for the insightful question. Please refer to W2.

---

> ### Author Response · Authors · 2025-11-22
> **Part II of the Response**
>
> > **Q2. How does Gaussian noise outperform more structured augmentations such as frequency- or context-based methods?**
>
>
> We thank the reviewer for the insightful comment. To address your concerns, we compare our personalized data augmentation with three typical methods: Fourier-based augmentation (FFT), Time warping (Warping) and Subsequence shuffling (Shuffling). 'FFT' transforms the time series into the frequency domain, perturbs its frequency components, and reconstructs the variant series. 'Warping' nonlinearly stretches or compresses the time axis. 'Shuffling' divides the series into consecutive segments and randomly rearranges their order.
> As shown in the following Table, the experimental results indicate that in long-tail imbalanced scenarios, our personalized augmentation strategy achieves the best performance. In contrast, subsequence shuffling performs the worst, as it may disrupt the inherent sequential structure, making it difficult for multi-scale analysis to accurately capture trend-related features.
>
> | Method                   | HAR-10 | HAR-50 | HAR-100 | ISRUC-10 | ISRUC-50 | ISRUC-100 | Epilepsy-10 | Epilepsy-50 | Epilepsy-100 |
> |---------------------------|--------|--------|---------|----------|----------|-----------|-------------|-------------|--------------|
> | FFT                       | 96.18  | 94.15  | 91.87   | 81.94    | 76.33    | 72.86     | 65.43       | 60.77       | 54.40        |
> | Warping                   | 96.37  | 93.60  | 91.65   | 81.31    | 74.25    | 71.87     | 63.32       | 57.68       | 53.23        |
> | Shuffle                   | 94.85  | 92.31  | 89.12   | 80.14    | 74.02    | 70.09     | 63.21       | 57.06       | 43.18        |
> | Ours| 97.41  | 94.99  | 92.01   | 82.77    | 77.71    | 73.12     | 65.81       | 61.04       | 55.41        |
>
>
>
> > **Q3. What role does the multi-scale, channel-independent design play in handling class imbalance?**
>
> We thank the reviewer for the insightful comment.
> **Multi-scale Strategy**. We adopt a multi-scale and channel-independent design to enhance feature representations for all classes, particularly tail classes. Compared to single-scale modeling, multi-scale analysis extracts features at different temporal resolutions, allowing tail samples to capture rich hierarchical information and obtain more informative representations.
> **Channel-independent Strategy**. Due to the limited number of tail samples, the channel-independent design ensures that key features in each channel are fully preserved, thereby improving the discriminative ability of tail classes. In contrast, channel mixing retains only similar elements and may discard important features of individual samples.
>
>
> > **Q4. The boundary-repulsion loss resembles margin-based or supervised contrastive objectives. Could the authors clarify its unique contribution and explain how it improves discrimination in long-tailed TSC?**
>
> We thank the reviewer for the insightful question. Conventional margin-based losses impose a fixed margin for all classes, requiring that each sample maintain a predetermined distance from the decision boundary, regardless of class or sample characteristics.
> In contrast, our approach does not enforce a uniform margin. Instead, it dynamically estimates the decision boundary for each pair of classes, encouraging the sample to move away from the boundary.
> This mechanism enhances tail-class discriminability in long-tailed TSC by effectively separating scarce tail samples from neighboring classes.

---

> > ### Author Response · Authors · 2025-11-22
> > **Part III of the Response**
> >
> > We sincerely thank Reviewer #sotr for the careful evaluation and constructive suggestions. Your insightful comments have enabled us to further refine the motivation and presentation of our work. We appreciate the time and effort you devoted to reviewing our manuscript.

---

> ### Comment · Reviewer_sotr · 2025-11-26
>
> Thank you for the authors’ response, which has addressed most of my concerns. However, I still find the integration of the three modules somewhat loosely connected. Given that a score of 8 corresponds to an accept decision, I believe the paper does not yet meet the threshold for direct acceptance. Therefore, I will maintain my original score.

---

> > ### Author Response · Authors · 2025-11-26
> > **Reply Rebuttal Comment by Authors**
> >
> > We thank the reviewer for your thoughtful review of our work. We sincerely appreciate your valuable feedback and your confidence in our contribution. Please feel free to contact us if you have any further questions. Thank you once again for your insightful suggestions and support.

---

### Author Response · Authors · 2025-12-03
**Summary of Rebuttal**

Dear Reviewers, ACs, SACs, and PCs,

We are sorry to hear about the recent OpenReview bug issue, and we fully support the proposed remedial actions.

At the same time, we would like to emphasize that we have always adhered to the guidelines and have not exploited the OpenReview bug. Thanks to the diligence and responsiveness of our reviewers, a portion of the meaningful discussions had already been completed before the OpenReview bug issue broke out. Although we addressed the further comments, the reviewers were no longer able to take further actions based on our responses.

To assist in the final assessment of our submission, we summarize the key contributions and innovations of our work below:

> **(1) Problem level.** We reveal that long-tailed time-series data exhibit higher inter-class similarity than images, leading to overlapping temporal features and severely degraded tail-class separability, which has been overlooked in prior work.

> **(2) Method level.** To address the challenge of feature similarity caused by data similarity, which hampers the learning of a robust feature space, we propose a novel framework, which includes three complementary strategies at the data, feature extraction, and optimization levels, with the third strategy being the most innovative. First, we propose a novel personalized augmentation strategy to effectively enhance data diversity and robustness for scarce classes. Second, we employ a multi-scale analysis approach to capture more informative patterns, such as trend information, thereby enhancing its ability to learn features of tail classes. **Finally, we propose a novel boundary-repelling regularization, which adaptively estimates decision boundaries based on the characteristics of the long-tailed data, and repels sample features from these boundaries to improve the separability of tail classes.**

> **(3) Long-tailed time series benchmark.** To facilitate future research and promote community development, we propose a comprehensive benchmark comprising multiple models and four datasets. Three of these datasets include three sub-datasets each, representing diverse long-tailed scenarios, while the fourth PM2.5 dataset is drawn from real-world data. Additionally, following the reviewer 5fq8’s suggestions, we extended our benchmark by further constructing linear and stepped versions to provide more diverse scenarios.

During the rebuttal phase, we addressed the reviewers' concerns through the following efforts:

> **(1) Additional Experiments:** We conducted extensive experiments to further validate the soundness of our model design and to provide a more comprehensive evaluation.

> **(2) Further Analyses:** We provided deeper analyses of both related work and the experimental results.

> **(3) Manuscript Refinement:** We  revised the manuscript to improve clarity for the readers.



In addition, we believe it is necessary to report the changes in our scores throughout the fruitful rebuttal phase before the OpenReview bug issue broke out.

| Reviewer | Interaction | Rating |
|----------|-------------|--------|
| sotr|The reviewer read our response and maintained the score. | 6|
| Xejq|No response received before the OpenReview bug issue broke out. | 6 |
| 5fq8|The reviewer responded, and we provided a further reply, but we were unable to continue the discussion.| 4 (no further discussion)|
| PqPE| The reviewer increased the score and raised further questions, to which we have responded. However, we were unable to continue the discussion.|4 (no further discussion)|
| Avg.     | - | 5 |

We sincerely appreciate the reviewer’s constructive feedback. In particular, Reviewer 5fq8 and PqPE raised additional questions and suggestions, demonstrating a strong willingness to help us further improve the quality of the paper. We have addressed these new questions and suggestions through additional experiments. Unfortunately, the OpenReview incident prevented any further discussion.

Once again, we sincerely thank all reviewers for their efforts in reviewing our paper and for maintaining active communication with us throughout the rebuttal period.

Best regards,

Authors

---

### Meta-Review · Area_Chair_Kojh · 2026-01-06

**Summary:**

This paper studies long-tailed time series classification, a practically relevant and challenging problem, and proposes a multi-scale feature space learning framework to improve robustness under class imbalance. Reviewers generally agree that the problem is well motivated and that the proposed method is technically sound. The use of multi-scale representations and tailored feature regularization is reasonable, and the experimental results show consistent improvements over several baselines on long-tailed benchmarks.

While the paper demonstrates solid incremental improvements, its conceptual novelty and broader impact remain limited. The proposed approach largely builds upon existing long-tailed learning and multi-scale feature learning techniques, and the paper does not clearly articulate a fundamentally new perspective or insight that advances the field beyond prior work. Moreover, although performance gains are reported, reviewers raised concerns that the improvements are often incremental and sensitive to design choices, with limited analysis of when and why the method substantially outperforms strong baselines. The paper would benefit from deeper theoretical insight or more compelling empirical evidence demonstrating generality across diverse long-tailed time series scenarios.

The rebuttal is constructive and addresses several reviewer questions, improving clarity and experimental completeness. Nevertheless, these improvements do not fundamentally change the overall assessment that the contribution is incremental rather than transformative, and therefore falls short of the level of significance expected at ICLR.

**Reviewer Concerns:**

Limited conceptual novelty beyond existing long-tailed and multi-scale learning approaches.

Empirical improvements are incremental and not always decisive over strong baselines.

Lack of deeper theoretical insight into why the proposed feature-space design improves long-tailed robustness.

The broader generality and impact of the method remain insufficiently demonstrated.

**Reviewer Scores:**

Reviewer scores were initially mixed. During the rebuttal phase, one reviewer revised their score upward from 2 to 4, reflecting that some concerns were partially addressed. However, no further score increases were made after that point, and other reviewers did not revise their scores. Based on the original reviews and the rebuttal, it is unlikely that the rebuttal would lead to sufficient upward score revisions to overcome the remaining concerns.

---

### Decision · Program_Chairs · 2026-01-26

Reject